# Investigating biomass burning aerosol morphology using a laser imaging nephelometer

Katherine M. Manfred[1,2], Rebecca A. Washenfelder[2], Nicholas L. Wagner[1,2], Gabriela Adler[1,2], Frank Erdesz[1,2], Caroline C. Womack[1,2], Kara D. Lamb[1,2], Joshua P. Schwarz[2], Alessandro Franchin[1,2], Vanessa Selimovic[3], Robert J. Yokelson[3], Daniel M. Murphy[2]

[1]Cooperative Institute for Research in Environmental Sciences, University of Colorado, Boulder CO, 80309, USA
[2]Chemical Sciences Division, Earth System Research Laboratory, National Oceanic and Atmospheric Administration, Boulder CO, 80305, USA
[3]Department of Chemistry, University of Montana, Missoula MT, 59812, USA

*Correspondence to*: Katherine M. Manfred (katherine.manfred@noaa.gov)

**Abstract.** Particle morphology is an important parameter affecting aerosol optical properties that are relevant to climate and air quality, yet it is poorly constrained due to sparse in situ measurements. Biomass burning is a large source of aerosol that generates particles with different morphologies. Quantifying the optical contributions of non-spherical aerosol populations is critical for accurate radiative transfer models, and for correctly interpreting remote sensing data. We deployed a laser imaging nephelometer at the Missoula Fire Sciences Laboratory to sample biomass burning aerosol from controlled fires during the FIREX intensive laboratory study. The laser imaging nephelometer measures the unpolarized scattering phase function of an aerosol ensemble using diode lasers at 375 nm and 405 nm. Scattered light from the bulk aerosol in the instrument is imaged onto a CCD using a wide-angle field-of-view lens, which allows for measurements at $4 - 175°$ scattering angle with ~0.5° angular resolution. Along with a suite of other instruments, the laser imaging nephelometer sampled fresh smoke emissions both directly and after removal of volatile components with a thermodenuder at 250°C. The total integrated aerosol scattering signal agreed with both a cavity ring-down photoacoustic spectrometer system and a traditional integrating nephelometer within instrumental uncertainties. We compare the measured scattering phase functions at 405 nm to theoretical models for spherical (Mie) and fractal (Rayleigh-Debye-Gans) particle morphologies based on the size distribution reported by an optical particle counter. Results from representative fires demonstrate that particle morphology can vary dramatically for different fuel types. In some cases, the measured phase function cannot be described using Mie theory. This study demonstrates the capabilities of the laser imaging nephelometer instrument to provide real-time, in situ information about dominant particle morphology, which is vital for understanding remote sensing data and accurately describing the aerosol population in radiative transfer calculations.

## 1 Introduction

Atmospheric aerosol particles absorb and scatter light, strongly influencing earth's climate (Davidson et al., 2005; Stocker et al., 2013). Size, composition, and morphology are key parameters in determining aerosol optical properties. Aerosol morphology is particularly important in implementing global radiative transfer calculations, since the directionality of scattered light from aerosol relative to the solar zenith angle will vary with particle shape (Kahnert and Devasthale, 2011). Most satellite and ground-based remote sensing retrievals assume that accumulation mode aerosol (~10 – 1000 nm) are spherical to determine quantities like size distribution and total loading based on measured solar reflectance or transmittance through the atmosphere (Holben et al., 1998; Levy et al., 2007; Omar et al., 2009). Improved models require more accurate information about particle morphology.

Biomass burning is a major source of aerosol, with average annual global emissions of 49 Tg (Stocker et al., 2013). Biomass burning particle emissions are composed predominantly of a mixture of fractal-like black carbon and spherical organic aerosol, along with a minor contribution from other species such as dust and inorganic salts (Adachi and Buseck, 2011; China et al., 2013; Hand et al., 2005; Li et al., 2003; Pósfai et al., 2004). Biomass burning and biofuel combustion accounts for 67% more black carbon particle emissions than fossil fuel combustion, and is also a major source of organic aerosol (Bond et al., 2004; Liousse et al., 1996). The complex morphology of biomass burning aerosol and its high atmospheric abundance make it an important target for improved characterization of optical properties.

In the past, morphology was generally inferred from dynamic shape factor experiments based on, for example, aerodynamic size and mass measurements (Hand and Kreidenweis, 2002; Slowik et al., 2004; Zelenyuk et al., 2006). These methods require multiple instruments, and are not well suited for measurements of evolving samples where high time resolution is desirable. Furthermore, the dynamic shape factor provides some information about the effective density of the particle, but does not provide important information on the optical properties of the particles. Alternatively, samples have been analyzed off-line using scanning or transmission electron microscopy (SEM/TEM) (Brodowski et al., 2005; McDonald and Biswas, 2004). Quantitative characterization using these methods relies on sophisticated software, can be quite time-consuming, and assumes that the morphology is not altered by deposition to the imaging substrate. LIDAR can be used to derive a scalar nonsphericity value based on the depolarization ratio of backscattered radiation. This value is used in remote sensing to indicate, for example, when mineral dust aerosol are present (Mishchenko and Sassen, 1998; Omar et al., 2009), but does not work well for accumulation mode particles (Murayama et al., 2003). Recently, a sophisticated method for determining the morphology of coarse mode aerosol using holographic imaging without interrupting the aerosol flow was demonstrated (Berg et al., 2017), however this method is unfeasible for accumulation mode particles.

The aerosol phase function is the angular distribution of the radiation scattered from the particles. It is dependent on the size distribution, complex refractive index, and morphology of the aerosol population. Instruments used for measurements of aerosol scattering phase function can be broadly grouped into four categories. The most common technique uses hemispheric measurements of backscatter and total scatter from integrating nephelometers (Anderson et al., 1996). Some nephelometers

also have an option to measure set angular regions by blocking portions of the forward scattered light (Chamberlain-Ward and Sharp, 2011). While these devices are reliable and robust, the measurements over large portions of the phase function do not provide much information about particle morphology due to the lack of angular specificity. Second, some studies use moveable detectors along a ring structure such that the phase function can be measured by sweeping the detector (or multiple detectors) across a broad range of angles (Holland and Gagne, 1970; Hovenier et al., 2003; Jaggard et al., 1981; Kuik et al., 1991; Perry

et al., 1978; Volten et al., 2001). While this allows for good angular coverage (up to $3° − 173°$ (Hovenier et al., 2003)), the method is generally bulky, must be mechanically stable, and requires a constant population of aerosol particles that does not change appreciably during the detector sweep. A third type of apparatus reported in the literature uses arrays of many photodetectors positioned along an arc, with some arrays including up to 36 separate detectors (Barkey et al., 1999; Gayet et al., 1997, 1998; Pope et al., 1992; West et al., 1997; Wyatt et al., 1988). This requires careful calibration of the detectors, and

generally suffers from low (~2°) angular resolution. The fourth category is the newest and features a pixel array detector with either an elliptical mirror (Curtis et al., 2007) or a wide-angle lens (Bian et al., 2017; Dolgos and Martins, 2014) used to image nearly all angles onto the array. This type of instrument uses a single detector with a simple, relatively compact design that does not require any moving parts and has great potential for field measurements that require higher time resolution. The first polarized imaging nephelometer used for field measurements employed a wide-angle lens and charge-coupled device (CCD)

to record the scattering phase function under two orthogonal linear polarization conditions at three wavelengths (473, 532, and 671 nm) and derive both the phase function and degree of linear polarization across the visible spectral region (Dolgos and Martins, 2014; Espinosa et al., 2017). This instrument was deployed on the NASA DC8 aircraft for the Deep Convective Clouds and Chemistry (DC3) campaign in 2012 (Barth et al., 2015) and the Studies of Emissions and Atmospheric Composition, Clouds and Climate Coupling by Regional Surveys (SEAC[4]RS) mission in 2014 (Toon et al., 2016).

Here we describe a modified commercial laser imaging nephelometer based on the same principles as the polarized imaging nephelometer that measures the unpolarized phase function at 375 and 405 nm. This method has the advantages of being online, real-time, and sufficiently sensitive for ambient aerosol loadings. We use this instrument to determine morphology characteristics of accumulation mode aerosol by imaging the aerosol phase function and directly measuring the effects of

morphology on radiative properties. This allows for a clearer understanding of how the dominant morphology of the measured aerosol sample affects, for example, radiative transfer calculations.

We deployed the laser imaging nephelometer as part of the Fire Influence on Regional and Global Environments Experiment (FIREX) intensive study at the Missoula Fire Sciences Laboratory to measure biomass burning aerosol. We compare the

retrieved phase functions to models of spherical and fractal particles to provide direct verification that the predominant morphology of the freshly emitted particles varied by fuel type, and demonstrate the ability to assign particle morphology from the phase function.

## 2 Experimental

### 2.1 Description of the laser imaging nephelometer

The laser imaging nephelometer (LiNeph; AirPhoton, Baltimore, MD, USA), shown schematically in Figure 1, measured nearly the entire unpolarized scattering phase function (~4 – 175°) of a bulk aerosol sample. Light at 375 nm and 405 nm was provided by two collimated, continuous wave diode lasers (LuxX; Omicron-Laserage Laserprodukte GmbH, Rodgau-Dudenhofen, Germany) with maximum operating powers of 70 mW and 120 mW respectively. These wavelengths were selected under the hypothesis that increased absorption by brown carbon particles in the UV compared to the blue may be observable in the measured phase functions, however we were unable to observe this under the experimental conditions. The output beams from the two lasers were co-aligned using a dichroic mirror before being steered through a quarter waveplate (to convert linear to circular polarization) and window into the sample chamber. The circular polarization of the beam allows retrieval of the unpolarized phase function, for a homogeneous sample (Bohren and Huffman, 1983), simplifies the experimental set-up by eliminating the need for a variable waveplate (Dolgos and Martins, 2014), and maximizes the data acquisition rate by only recording the phase function under one polarization condition at each wavelength. A set of three apertures (4 mm diameter) blocked stray light before the beam passed through the aerosol sample. The two beams traveled the same beam path inside the chamber, with collimated diameters of approximately 3 mm, and exhibited Gaussian intensity profiles (see Figure S4). A beam dump minimized light back-reflected through the sample. The lasers were modulated on and off sequentially. The total data acquisition time for each cycle (with 1 s integration at each wavelength and readout of two CCD images) was 5 – 6 s, equivalent to a duty cycle of ~20% for each wavelength.

A wide angle field-of-view fish-eye lens (FE185C046HA-1; Fujifilm, Tokyo, Japan) collected the light and imaged it onto a 16-bit, $2750 \times 2200$ pixel CCD array detector cooled to -40 °C (Trius-SX694; Starlight Xpress, Bracknell, UK) with a very low dark current and 1 s integration time. The combination of a wide-angle lens and CCD array allow imaging of the phase function for aerosol and gas molecules within the volume of the beam, as discussed in Dolgos and Martins (2014). The CCD combined individual pixels into $2 \times 5$ pixel bins, where the axes correspond to those transverse and parallel to the laser beam respectively. A sample image is shown in Figure 2. The lasers and CCD were controlled by a custom LabVIEW program that also acquired auxiliary measurements from the mass flow controllers and pressure sensor (see Section 2.3), as well as the laser power measured at the diode heads, for each image.

## 2.2 Image processing for the laser imaging nephelometer

The raw CCD images were processed in five steps to produce an aerosol scattering phase function by correcting for the CCD dark background, scattered light due to internal surfaces, and Rayleigh scattering. First, a dark background spectrum was subtracted from each CCD image. This is necessary because CCD detectors produce non-zero dark current in the absence of light due to thermal noise and an electronic offset. The dark background image was acquired with the same integration time as the aerosol images and was measured before each fire. Second, light scattering from surfaces within the sample volume was removed by subtracting a laser power-normalized correction from each image. The scattering correction was determined by filling the chamber with helium, which has a negligible scattering coefficient, and taking an average of ~20 images at each wavelength in order to account for light reaching the CCD from the instrument body and optics. Third, each strip of the corrected image was integrated and an additional constant offset due to temperature variations in the CCD pixel array was removed. Fourth, the Rayleigh scatter contribution of the dry air, which was measured before each experiment, was pressure-corrected and removed from the signal to determine the particle-only scattering. Finally, an angle-dependent correction factor was applied. This correction factor was determined daily using Rayleigh scattering from dried and filtered ambient air, and Rayleigh scattering coefficients for dry air from the literature (Bodhaine, 1979; Penndorf, 1957). Since the air was dried and scrubbed, corrections for additional water vapor or $NO_2$ were not needed; likewise, corrections for higher modern-day levels of $CO_2$ were deemed unnecessary since additional $CO_2$ (compared to "standard air" in the 1950s (Penndorf, 1957)) would result in a maximum error of 0.03% in the Rayleigh scattering cross-section. The consistency of the calibration factor from day-to-day (<3% variation) implies that the lens, window, and chamber walls did not accumulate particles or become dirty over time.

The wavelength-dependent pixel-to-angle calibration was measured in the laboratory before and after the FIREX measurements using standard monodisperse polystyrene latex spheres (PSLs) with diameters ranging from 100 to 895 nm. For these experiments, a solution of monodisperse NIST-traceable PSL (3000 Series Nanospheres, ThermoFisher Scientific, Waltham, MA, USA) in pure water was atomized, dried, and size-selected by a differential mobility analyzer (DMA) before being sampled by the imaging nephelometer and a condensation particle counter (CPC 3022; TSI Inc., Shoreview, MN, USA). The measured phase functions were compared to modelled phase functions from Mie theory for each monodisperse samples of PSLs used to determine a wavelength-dependent angular calibration. There was a small difference in the calibrations at 375 and 405 nm due to achromatic behavior of the wide-angle lens. The pixel-to-angle relationship was found to be linear (see Supplementary), with each pixel bin corresponding to ~0.5° scattering angle. The laser power in the chamber was measured using a power meter (Gentec-EO, Quebec City, Quebec, Canada) and calibrated against the diode head laser power reported by the laser to account for transmission losses and finite aperture size.

The total integrated aerosol scatter was determined for each image by integrating the illuminated area and applying a size-dependent truncation using the measured aerosol size distribution and assuming spherical particles. While the total scattering measured at 375 nm is greater, the overall shape of the phase functions at 375 and 405 nm are quite similar within the signal level of the measurements. Therefore only phase functions measured at 405 nm will be shown, and we focus on the analysis of the 405 nm phase functions here.

## 2.3 Aerosol sampling at the Fire Sciences Laboratory

The Missoula Fire Sciences Laboratory is a United States Department of Agriculture (USDA) facility with a large (~3000 m$^3$) combustion laboratory where controlled burns are studied. During the FIREX 2016 intensive laboratory study, natural fuels common to western US wildfires were burned and measured by a wide array of instrumentation. For the fires studied here, the fuel bed was lit and combustion laboratory closed off to allow the smoke to become well-mixed, with 3 – 4 hours of sampling time following 15 min of combustion. Similar sampling conditions at the Fire Sciences Laboratory have been described previously (McMeeking et al., 2009).

Biomass burning aerosol was sampled from the smoke-filled room through ~30 m of 6.4 mm OD copper tubing connected to a sampling inlet shared by a suite of aerosol optical and composition instruments, as shown in Figure 3, located in an adjacent room. A 2.0 volumetric L min$^{-1}$ (vlpm) aerosol sample flow passed through an impactor with a 50% cut point at 1 μm, then was dried and scrubbed of O$_3$ and NO$_x$. 1.0 vlpm of sample flowed through a thermodenuder at 250°C with an activated carbon lining to remove the evaporated gases, while 1.0 vlpm of sample flowed through an unheated bypass channel, controlled using an automated valve. It was then diluted with dried, filtered, and scrubbed (O$_3$ and NO$_x$) compressed ambient air in a ~1 L mixing volume. The dilution ratio of the original smoke sample was typically about 50:1.

A collection of aerosol instruments sampled from the mixing volume, including the imaging nephelometer, an integrating nephelometer, and cavity ring-down photoacoustic spectrometer (CRD PAS) (Lack et al., 2012; Langridge et al., 2011). A laser aerosol spectrometer (LAS; TSI Inc., Shoreview, MN, USA) (Jonsson et al., 1995) periodically measured the aerosol size distribution from the mixing volume during each fire. Likewise, a single particle soot photometer (SP2; Droplet Measurement Technologies, Longmont, CO, USA) measured the concentration of refractory black carbon aerosol, and provided an estimate of the enhancement of light absorption due to lensing by internally mixed material coating refractory black carbon cores (Schwarz et al., 2006, 2008).

The laser imaging nephelometer sampled in series with a traditional commercial integrating nephelometer (3563; TSI Inc, Shoreview, MN, USA). This instrument, which is commonly used in laboratory and field studies, contains a halogen lamp to illuminate the sample and measures the total scatter and backscatter at red (750 nm), green (550 nm), and blue (450 nm) wavelengths using three different photomultiplier tubes with narrow bandpass filters (Anderson and Ogren, 1998;

Heintzenberg and Charlson, 1996). The flow through the nephelometers was 10 – 15 slpm, controlled by a mass flow controller (Alicat Scientific, Tucson, AZ, USA) and diaphragm pump (Gast Manufacturing, Benton Harbor, MI, USA). The integrating nephelometer was upstream of the imaging nephelometer and a baratron (MKS, Andover, MA, USA) recorded the pressure at the outlet of the imaging nephelometer.

During some fires, aerosol particles were collected on silicon wafers for off-line analysis using a scanning electron microscope (SEM). These were collected on a separate line from the general inlet with a 1 μm 50% cut-point impactor and a similar option to go through an identical thermodenuder at 250°C or unheated bypass channel. The sample flow was 1 slpm through a cascade impactor (Mini-MOUDI; MSP, Shoreview, MN, USA). The substrates were attached to the impactor plate below the ~100 nm

nominal cut-point stage. After collection, the silicon chips were attached to SEM stubs with conductive adhesive, and later sputter coated with up to 5 nm of platinum to prevent damage from charging in the SEM. The samples were analyzed under high vacuum with accelerating voltages of 5 – 15 kV using an SEM (SU3500; Hitachi, Tokyo, Japan) at the University of Colorado Nanofabrication Characterization Facility.

## 3 Data Analysis

### 3.1 Parameterizations of phase functions from particle morphology

Biomass burning aerosol consist predominantly of a mixture of primary fractal-like black carbon particles and spherical organics formed by condensation of semi-volatile species from the gas phase (Adachi and Buseck, 2011; Hand et al., 2005; Li et al., 2003; Pósfai et al., 2004). In this paper, we adopt the definition of black carbon provided by Bond et al. (2013): "a distinct type of carbonaceous material that is formed primarily in flames, is directly emitted to the atmosphere, and has a

unique combination of physical properties." To look at how morphology affects the retrieved phase function, we used two morphology parameterizations: Mie theory for spherical particles and Rayleigh-Debye-Gans (RDG) for fractal-like particles. The spherical particles are assumed to be predominantly composed of organic matter, whereas black carbon particles formed by incomplete combustion are fractal-like. Since fresh (< 4 hrs) emissions were measured in the absence of sunlight, it is unlikely that significant oxidative aging occurred in the smoke chamber. The smoke chamber relative humidity was less than

40% for all fires measured, so we assume there was no significant restructuring of the black carbon agglomerates by organic or sulfuric acids (Xue et al., 2009; Zhang et al., 2008).

Mie theory describes electromagnetic radiation scattering by spherical particles with size parameters ($x = \pi d/\lambda$) of approximately unity, indicating that the diameter of the sphere ($d$) is on the same length scale as the wavelength of the radiation

($\lambda$) (Bohren and Huffman, 1983). Rayleigh-Debye-Gans (RDG) theory has been used extensively to approximate the physical properties of fractal-like aerosol (Chakrabarty et al., 2007; Liu et al., 2013a; Sorensen, 2001). The approximation assumes that fractals are composed of small identical monomers, and requires the following conditions to be fulfilled: $|m - 1| \ll 1$ where

$m$ is the complex refractive index; and $|2x_P(m-1)| \ll 1$ with $x_p$ equal to size parameter of the monomer. For the parameterization of biomass burning fractal-like aerosol particles used here, these requirements are adequately met (Farias et al., 1996). The model assumes negligible interactions between the particles themselves, and several studies have verified that multiple scattering effects are indeed negligible for fractal-like aerosol (Farias et al., 1996; Liu et al., 2013a).

Fractal clusters are defined by the following parameterization (Forrest and Witten, 1979; Sorensen, 2001):

$$N_s = k_o \left(\frac{R_g}{a}\right)^{D_f} , \tag{1}$$

where $N_s$ is the number of monomer spherules per agglomerate, $R_g$ is the radius of gyration, $a$ is the monomer radius, $k_o$ is the fractal prefactor, and $D_f$ is the fractal dimension. Many studies have used theoretical models and experimental measurements

to determine appropriate $D_f$ and $k_o$ values for aerosol produced by hydrocarbon combustion (e.g. Sorensen and Roberts, 1997). The fractal dimension can physically range from 1 (for highly branched structures) to 3 (for clumped structures). Generally, the fractal dimension is determined to be around $1.75 - 1.85$ for fresh (non-collapsed) soot aerosols produced by diffusion-limited aggregation (i.e. the probability of colliding particles to stick together is unity) (Brasil et al., 2000). There is, however, a wide range in the reported fractal prefactor, which is dependent on the packing and overlap between monomers. Prefactors

as high as 8 have been reported (Köylü et al., 1995), though most values reported in the literature for fossil fuel combustion are between $1 - 2$ (Brasil et al., 2000; Sorensen, 2001; Sorensen and Roberts, 1997).

While fractal-like particles produced from fossil fuel combustion have been studied extensively, few systematic measurements of the fractal parameterization of fresh biomass burning combustion products have been undertaken. Chakrabarty et al. (2006)

reported $D_f$ values in the range of $1.67 - 1.83$ and $k_o$ values from $2.05 - 2.90$ for varied biomass fuels. Gwaze et al. (2006) measured $k_o = 2.77$ and $D_f = 1.83$ for biomass burning aerosol from wood, although the authors note that 1.83 is a lower limit on $D_f$. In this work, unless otherwise noted, $k_o$ and $D_f$ are taken from Chakrabarty et al. (2006) based on fuel type. The scattering angle-dependent structure factor, which is needed to calculate the RDG phase function, was adopted from Sorensen et al. (Liu et al., 2013a; Sorensen et al., 1992; Yang and Köylü, 2005). Details of the phase function calculations based on the

RDG model can be found in Liu et al. (2013a) and Kandilian et al. (2015).

We used measured size distributions from the LAS to predict phase functions for spherical and fractal-like biomass burning particles and compared those with measurements by the imaging nephelometer. The LAS measures particle size from the amount of light scattered by a 633 nm laser beam. The LAS size bins had been calibrated using nominally spherical ammonium

sulfate particles ($m = 1.53$ at 633 nm) (Haynes, 2013). Equivalent fractal sizes were defined for each LAS size bin by varying the number of spherules per agglomerate (with constant monomer diameter, fractal prefactor, fractal dimension, and refractive index) to match the total scattering within the measured angle range ($33 - 75°$ and $105 - 147°$) defined in the manufacturer's

specifications (see Supplementary for more information). For RDG model calculations, the monomer diameter ($2a$) was assumed to be 50 nm, based on SEM images from multiple fires and in agreement with past studies of biomass burning (Chakrabarty et al., 2006; Gwaze et al., 2006). A wavelength-independent refractive index for black carbon ($m = 1.95 + 0.8i$) was assumed (Bond and Bergstrom, 2006). The values of $D_f$ and $k_o$ were taken from the literature for similar fuel types and are detailed below (Chakrabarty et al., 2006). While the phase functions modeled using RDG are sensitive to these values, as well as the monomer diameter, a comprehensive analysis of the full range of possible RDG parameterization is beyond the scope of this paper; rather, here we present an initial "best guess" of the RDG parameterization based on the limited literature on biomass burning-produced soot and SEM micrographs. Spherical particles were assumed to be predominantly organic material, and a representative refractive index for humic-like substances was assumed at each wavelength ($m = 1.64 + 0.12i$ at 375 nm; $m = 1.64 + 0.11i$ at 405 nm) (Dinar et al., 2008; Hoffer et al., 2005). A comparison of Mie theory calculations for a range of refractive indices is available in the Supplementary.

## 4 Results and Discussion

### 4.1 Precision and accuracy of the scattering phase function

The precision of the imaging nephelometer was quantified using an Allan-Werle variance analysis (see Figure S2 in Supplementary) to the integrated signal for individual angle bins (Allan, 1966; Werle et al., 1993). For particle-free air, the standard deviation at 5° scattering angle (~0.5° angle bin) was $6.9 \times 10^{-6}$ m$^{-1}$ for 1 min and decreased to $1.6 \times 10^{-6}$ m$^{-1}$ for 10 min. At 90° (~0.5° angle bin), where Rayleigh scattering is at a minimum, the standard deviation is $5.9 \times 10^{-6}$ m$^{-1}$ for 1 min and $8.7 \times 10^{-7}$ m$^{-1}$ for 10 min.

The two main sources of uncertainty for aerosol measurements are variations in the number density and size of particles in the laser beam, and background scatter attributed to temperature variability in the CCD. Errors arising from fluctuations in the particles, notably for large particles, crossing the beam can be reduced by averaging the signal over long periods. To compare with the uncertainty for gas phase measurements above, the standard deviation with 1 min averaging for a sample of 600 nm PSL at a concentration of 23 cm$^{-3}$ is about $1.7 \times 10^{-5}$ m$^{-1}$ at the 5° scattering angle bin. Given the beam volume of approximately 12.5 cm$^3$, there would be on average 290 aerosol particles measured across the full angle range. While the particles measured in the biomass burning samples were mainly smaller (~200 – 300 nm) with higher number densities, this demonstrates the importance of averaging when measuring large particles and lower concentrations in order to reduce statistical variability. In contrast, errors arising from the background signal are more systematic and difficult to quantify. The subtraction procedure outlined in Section 2.2 was implemented to minimize this error. Uncertainty on the calibration curve to convert pixel intensity to scattering is less than 3%.

## 4.2 Comparison of integrated scattering

The laser imaging nephelometer sampled aerosol from 11 separate fires using the experimental configuration described above. In this analysis we focus on two representative fires: pine (Fire A) and dry sagebrush (Fire B). These fires were chosen qualitatively to juxtapose phase function measurements for two classes of fuel type. For Fire A (#086), the fuel source was a mixture of dead wood (logs) of different diameters, litter, duff, and canopy branches with needles from lodgepole pine (*Pinus cortata*), while for Fire B (#085) the fuel source was solely comprised of dry sagebrush branches (*Artemisia tridentata*). The fuels were gently dried prior to burning, with moisture content <12%. Both fires exhibited a combination of flaming and smoldering processes, based on video observations of the fuel bed. The modified combustion efficiency (MCE), defined as $CO_2 / (CO_2 + CO)$, was 0.949 for Fire A and 0.938 for Fire B, and therefore both can be characterized as "flaming-dominated" (Reid et al., 2005). These MCE values were time-averaged over the full combustion period. The time evolution of the MCE, and thereby information about the impact of flaming and smoldering on aerosol optical properties during stages of combustion, was lost due to the mixing of gases and aerosol in the combustion laboratory (Liu et al., 2013b; Pokhrel et al., 2016). The effect of MCE on aerosol optical properties is discussed in other FIREX studies (Liu et al., 2013b; Pokhrel et al., 2016; Selimovic et al., 2017).

The accuracy of the total scattering measured by the imaging nephelometer is demonstrated by comparison with two independent measurements of aerosol scattering that were taken simultaneously. Figure 4 (a) shows the total scatter measured by the imaging nephelometer at 405 nm compared with the integrating nephelometer and CRD PAS for Fire A. The scattering Ångström exponent (SAE) measured between 450 and 550 nm by the integrating nephelometer was used to extrapolate scatter at 405 nm for this instrument. The total scattering measured by the CRD PAS was determined by subtracting the photoacoustic absorption value, measured at 401 nm, from the cavity ring-down extinction at 405 nm. In all cases, measurements have been corrected for dilution in the shared inlet, and the retrieved scattering represents that of the original smoke sample. The good agreement between all three measurements provide confidence in the total scatter retrieved by the imaging nephelometer (see Figure S1 in Supplementary for correlation plot).

The square-wave appearance of the scatter during the experiment is a result of alternating between the thermodenuder and unheated bypass channels on the shared inlet. The large decrease in total signal when the aerosol was denuded (~90% reduction) indicates that, for this fire, volatile organic material accounts for the majority of the light scattering at 405 nm. This is confirmed by the volume distribution measurements taken from the LAS, indicating that the total mass and average diameter decrease after the thermodenuder (see Figure 5(a)). The SP2 provides an absorption enhancement estimate by applying Mie theory with an assumed value for the index of refraction of internally-mixed material ($n = 1.45$) and a coated-sphere morphology (Schwarz et al., 2008) to measurements of the light scattered before significant volatile coating vaporization (Gao et al., 2007). The enhancement is reported to be as high as 1.6 for the undenuded sample, which suggests significant volatile

coatings on non-volatile cores (see Table 1). The absorption Ångström exponent (AAE) reported by CRD PAS between 401 and 532 nm is between 2.0 – 2.7, which is in agreement with literature values (>1 and <4) for biomass burning aerosol containing organic and black carbon material (Bergstrom et al., 2007).

In contrast, Figure 4 (b) shows that Fire B produced a more significant fraction of non-volatile aerosol indicated by the total scattering signal after the thermodenuder being only about 60% less than after the bypass channel. The CRD PAS and imaging nephelometer agree well throughout this experiment. The higher scattering retrieved by the integrating nephelometer is likely due to errors arising from extrapolating from 450 nm. The integrating nephelometer data end at 13:58 because of an automatic recalibration. Interestingly, for Fire B the volume size distribution as measured by the LAS is much narrower and does not
appear to shift significantly for the denuded products as was seen in the previous example. This could arise if the aerosol population has a predominantly fractal character. The side-scattering at the angles measured by the LAS is much less size-dependent in the RDG model compared to Mie calculations, and therefore the LAS distribution would be expected to narrow for fractal-like particles. The SP2 absorption enhancement estimate is significantly lower for this fire (1.1 – 1.2), indicating very little coating on the refractory black carbon particles. In addition to the high fraction of scatter from the nonvolatile
components, the AAE for this fire is much closer to unity, indicating absorption proportional to light frequency such as occurs for black carbon (see Table 1) (Bergstrom et al., 2002).

**4.3 Determination of biomass burning aerosol morphology from model parameterizations**

Figure 6 shows measured phase functions at 405 nm averaged over one individual cycle of the thermodenuder and bypass channel for well-mixed smoke from Fire A. The measurements are overlaid with the model predictions assuming spherical
particles representative of organic aerosol (Mie theory) and fractal-like black carbon particles (RDG theory) based on the LAS particle size distribution. For aerosol measured after the bypass channel (Figure 6(a)), the Mie theory curve matches the measurement reasonably well. Since the dominant aerosol type are highly volatile (i.e. evaporated at $\leq 250°C$), they are likely to be semi-liquid organics that will readily form spherical particles and be well-represented by Mie theory. A comparison of phase functions assuming different refractive indices and fractal parameterizations is available in the Supplementary. In
contrast, the phase function of the denuded products looks much more similar to the RDG fractal model prediction based on the ponderosa pine parameterization ($k_o = 2.32; D_f = 1.69$) from Chakrabarty et al. (2006) (Figure 6(b)). This would suggest that the refractory material has a predominantly fractal-like morphology that is well described using this parameterization for fresh biomass burning emissions.

This implication from the phase function measurements is further supported by SEM images (Figure 7), which show that the non-volatile (denuded) components are mostly fractal-like particles. Using the SEM software, the average monomer diameter was measured for monomers clearly visible on the outer edges of the agglomerates; the average monomer diameter (excluding Pt coating) is 50 ± 10 nm. A simple box-counting fractal analysis was performed using image analysis software (ImageJ,

(Image Processing and Analysis in Java), National Institutes of Health) on the fractal-like particles measured from Fire A (thermodended) to estimate the fractal dimension (Karperien, 2013). This algorithm calculates a fractal dimension ($D_{fB}$) based on the relationship between the length scale ($\epsilon$) and number of boxes containing a portion of the fractal at each scale ($N_\epsilon$) (Theiler, 1990):

$$D_{fB} \approx \frac{\log(N_\epsilon)}{\log(\epsilon)}. \tag{2}$$

The fractal dimension retrieved (1.87 ± 0.06) likely represents an upper limit as the Pt coating on the particles will increase apparent overlap of monomers. Both the monomer size and fractal dimension (considering it is an upper limit) are in line with the literature values used for the RDG models based on previous studies of biomass burning particles (Chakrabarty et al., 2006; Gwaze et al., 2006). Due to the coating, we are not able to retrieve the fractal dimension and prefactor using other methods previously demonstrated for fractal-like aerosol particles (Brasil et al., 1999; Chakrabarty et al., 2006). Additionally, it is unclear whether any deformation of the fractal-like agglomerates occurred upon deposition to the silicon substrate. Interestingly, there also appear to be a small fraction of spherical particles in the SEM micrographs that could fall into the classification of "tar balls" – amorphous, non-volatile organic aerosol previously observed in biomass burning aerosol populations that survive thermal denuding (Adachi and Buseck, 2011; Chakrabarty et al., 2010; China et al., 2013; Hand et al., 2005; Pósfai et al., 2004).

Figure 8 shows phase function plots for Fire B, which, in contrast to Fire A, exhibits a significantly higher fraction of nonvolatile aerosol. In both cases, the measured phase function is not described well using Mie theory, exhibiting significantly higher scatter at angles < 30°. The LAS size distribution would have to be shifted to higher diameters by a factor of over 2.5 to predict a spherical phase function with the observed forward scatter behavior seen in Figure 8(a); in this case, however, the model would overestimate total scatter by a factor of ten. The phase function measured for the total aerosol population (Figure 8(a)) shows good agreement with the RDG model, assuming the parameterization for sage fuel ($k_o = 2.56; D_f = 1.79$) from Chakrabarty et al. (2006). The agreement is less good for the thermodenuded aerosol (Figure 8(b)), where the biomass burning RDG parameterization underestimates the measured forward scattering. In this case, a second RDG-modelled phase function is overlaid to demonstrate how the fractal phase function varies with the selected parameterization; this second case assumes a parameterization typical of fossil fuel combustion products measured in the laboratory ($k_o = 1.2; D_f = 1.75$) (Sorensen, 2001; Sorensen and Roberts, 1997). The measured phase function appears to lie between the two RDG parameterizations, which implies that the fractal dimension and fractal prefactor for the denuded (non-volatile) aerosol are best represented by lower values than applied to the bypass channel scenario. Since the agglomeration process for the aerosols is the same, changes undergone because of the thermodenuder must account for the observed difference between the two cases. We hypothesize that a thin layer of organic coating may have been removed by the thermodenuder, resulting in less overlap between monomers. This would lead to a decrease in $k_o$ (less tight packing), while leaving $D_f$ largely unchanged (Chakrabarty et al., 2007; Gwaze et al., 2006). A study of fractal-like agglomerates produced by a diesel engine similarly measured no significant change to the

fractal dimension after removing volatile coatings with a thermodenuder (450°C) (Skillas et al., 1998). Additionally, soot agglomerates emitted by ethylene and methane flames did not exhibit restructuring after being heated to $250 - 270°C$ (Bhandari et al., 2017). If the fractal-like particles underwent restructuring due to heating, one would expect that the fractal dimension ($D_f$) would increase due to partial collapse of the structure. However, the shape of the measured phase function is qualitatively

in agreement with models assuming fractal dimensions of $1.75 - 1.79$ (Figure 8(b)), suggesting that the fractal dimension does not change and, consequently, no significant restructuring occurs in the thermodenuder. Unfortunately, no samples were collected for SEM analysis from this fire.

While the present study focused solely on fresh biomass burning emissions, it highlights that assuming similar optical

properties (Mie theory) for all biomass burning aerosol may be inappropriate. Many remote sensing platforms assume spherical morphology for biomass burning aerosol (Dubovik et al., 2000; Omar et al., 2009; Remer et al., 1998). This study indicates that this is appropriate for the aerosol with high volatile organic content, but leads to highly inaccurate assumptions of the phase function for the dry brush fuel example, which is likely to have a higher contribution from black carbon. Since this work is primarily intended to demonstrate the capabilities of the instrument, we refrain at this point from drawing definite

conclusions about atmospheric biomass burning aerosol from these two examples from controlled burns. It would be useful to collect more information about how the morphology of fresh emissions evolves with aging in the atmosphere. Studies have shown that biomass burning aerosol are coated with organic material within hours in the atmosphere (e.g. Akagi et al., 2012; Vakkari et al., 2014). However, if there are conditions under which fractal-like particles do not immediately collapse or accumulate sufficient organic coatings to become spherical, then remote sensing retrievals of wildfire plumes from dry brush

or grasses may significantly underestimate the forward scatter from the aerosol. Cheng et al. (2013) showed recently that assuming spherical morphology for simulated fractal-like soot particles leads to a significant underestimation in the aerosol absorption, as well as a large error in the scattering phase function, for remote sensing platforms that use reflectance. This error would likely have been unimportant for past retrievals from MODIS, for example, since cloud masking algorithms often misclassified thick smoke (typically fresh plumes) as clouds (Giglio et al., 2016). However, with improved biomass burning

cloud masking algorithms, it will be interesting to see if MODIS retrievals of thick, fresh smoke plumes will be accurate with a spherical morphology algorithm. This will affect not only the accuracy of size distribution and total loading retrievals, but may also be important for radiative transfer models. Due to the more significant forward scatter component for fractal particle scattering phase functions, it is possible that ground-based scanning radiometer retrievals, such as the Aerosol Robotic Network (AERONET), overestimate the contribution from coarse mode particles in order to account for high measured forward scatter

using Mie theory. This biasing of the size distribution of biomass burning aerosol would also have important repercussions for estimates of radiative forcing.

# 5 Summary and conclusions

This study demonstrates that a new laser imaging nephelometer can provide real-time, online phase function measurements of an accumulation-mode aerosol sample. The total scattering values retrieved compare well with both a traditional integrating nephelometer and with an extinction-minus-absorption method. In addition to total scatter, the ability to directly measure scattering phase function provides further information about the optical properties of the bulk aerosol that can be used to determine dominant particle morphology. Previously, particle morphology was inferred using a combination of aerodynamic and mobility measurements of particle diameter, or was measured off-line using transmission or scanning electron microscopy. This new instrument offers a more convenient, quantitative method to determine the radiative effects of varying morphologies. While we demonstrated here the potential inaccuracy of assuming spherical scattering behavior for all biomass burning aerosol, this method also has potential for direct retrievals of phase functions for other non-spherical particles, such as mineral dusts, pollens, and ice crystals. Direct, accurate, in situ measurements of bulk aerosol phase functions offer new, important ways of improving the accuracy of both remote sensing retrievals and radiative transfer calculations.

## Acknowledgements

This project was supported by NOAA and the NASA Radiation Sciences Program. The authors would like to thank Karl Froyd, Chuck Brock, and Bernie Mason for useful discussions, as well as Dan Cziczo for lending us the Mini-MOUDI impactor. We are grateful to Jim Roberts and Carsten Warneke for organizing the FIREX intensive in Missoula.

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

**Table 1: Selected measured values demonstrating the key differences between the aerosol products from Fire A and Fire B. Note: TD = thermodenuded**

| Parameter | Fire A | Fire B |
|---|---|---|
| *AAE (401-532 nm)* | 2.0 – 2.7 | 1.4 – 1.6 |
| *SP2 absorption enhancement estimate* | 1.4 – 1.6 (bypass) <br> 1.12 – 1.14 (TD) | 1.17 – 1.20 (bypass) <br> 1.11 – 1.12 (TD) |
| *Volume size distribution peak (nm)* | 250 (bypass) <br> 200 (TD) | 250 (bypass) <br> 230 (TD) |
| *Fraction total scatter TD at 405 nm* | ~0.1 | ~0.4 |
| *Fuel type* | Lodgepole pine | Sagebrush |

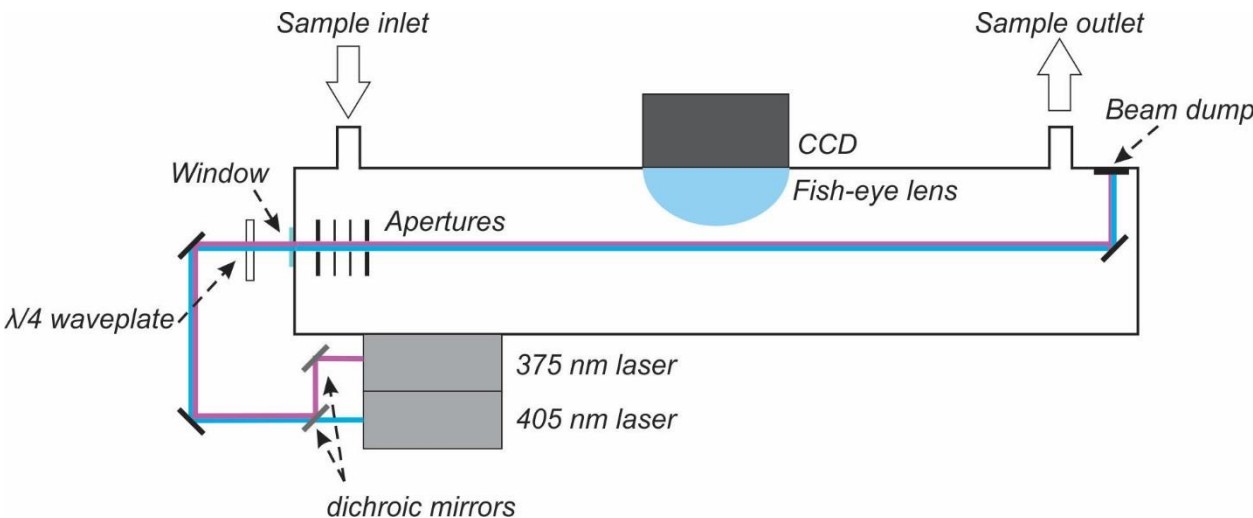

**Figure 1: Laser imaging nephelometer instrument diagram showing beam paths of the 375 and 405 nm beams, which are cycled in sequence in the experiment.**

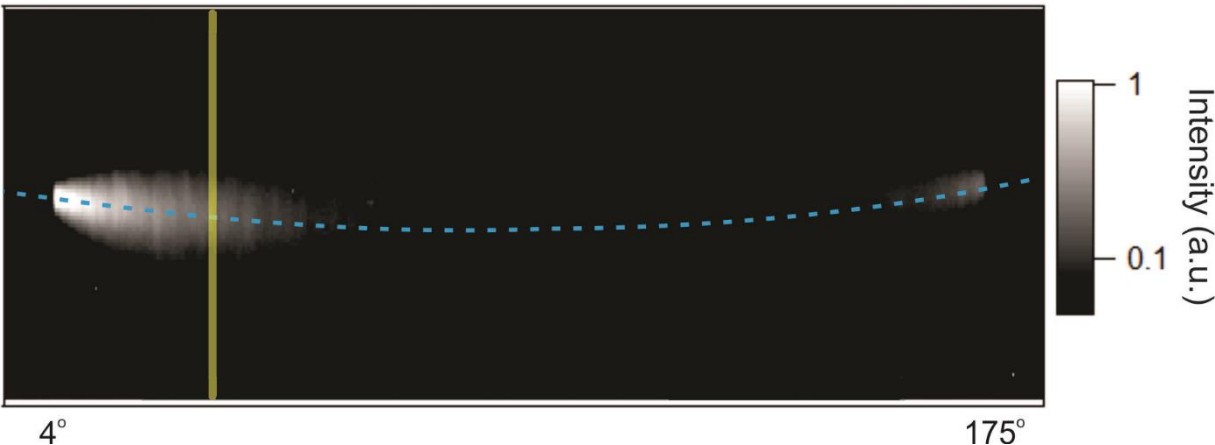

**Figure 2: Example raw CCD image during Fire A for the 405 nm laser. The yellow rectangle indicates the direction across which the image is integrated in order to determine scattering for a single angle bin (~0.5°). The image represents 100 × 400 pixel bins (transverse × parallel to laser beam). The blue dashed line shows the center of the laser beam; the slight curvature arises from the wide-angle lens being slightly off-center relative to the laser path.**

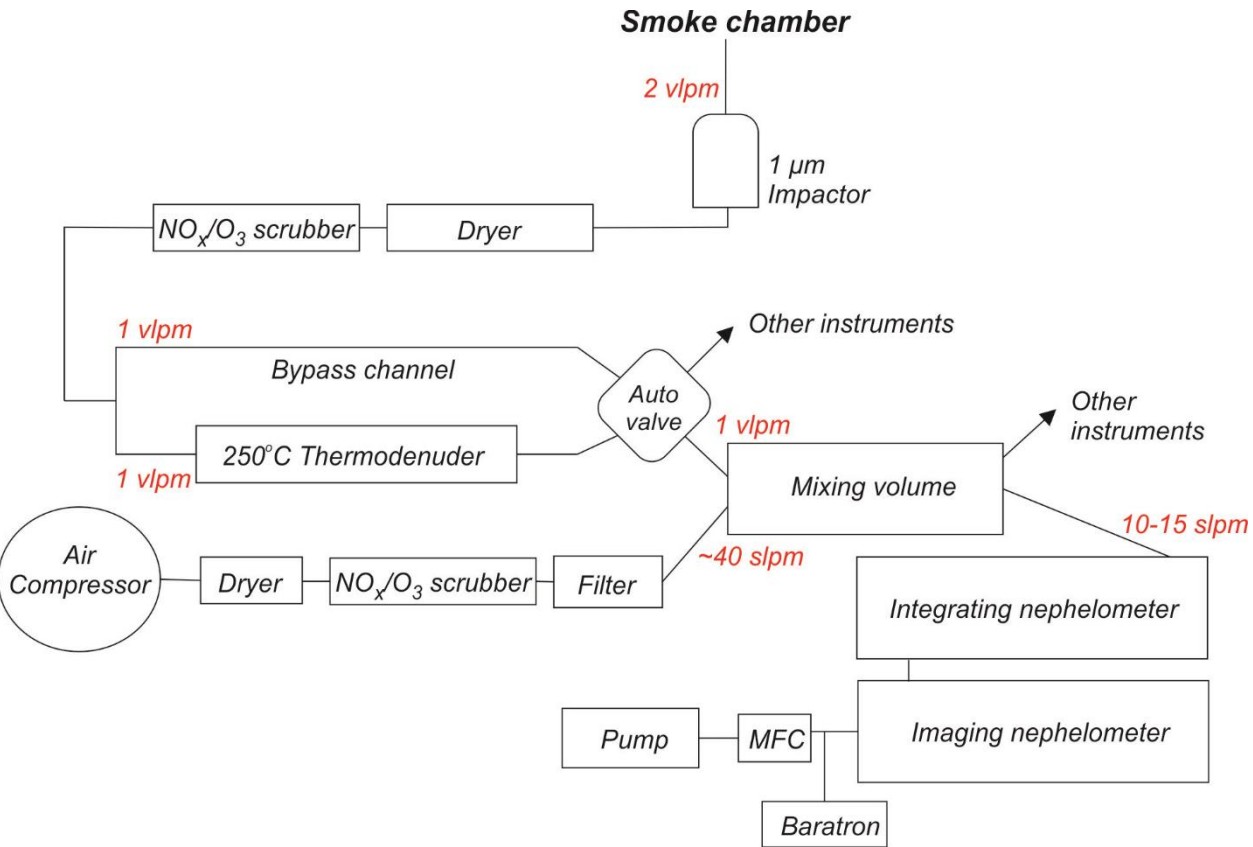

**Figure 3: Schematic of NOAA aerosol sampling inlet at Missoula Fire Sciences Laboratory.**

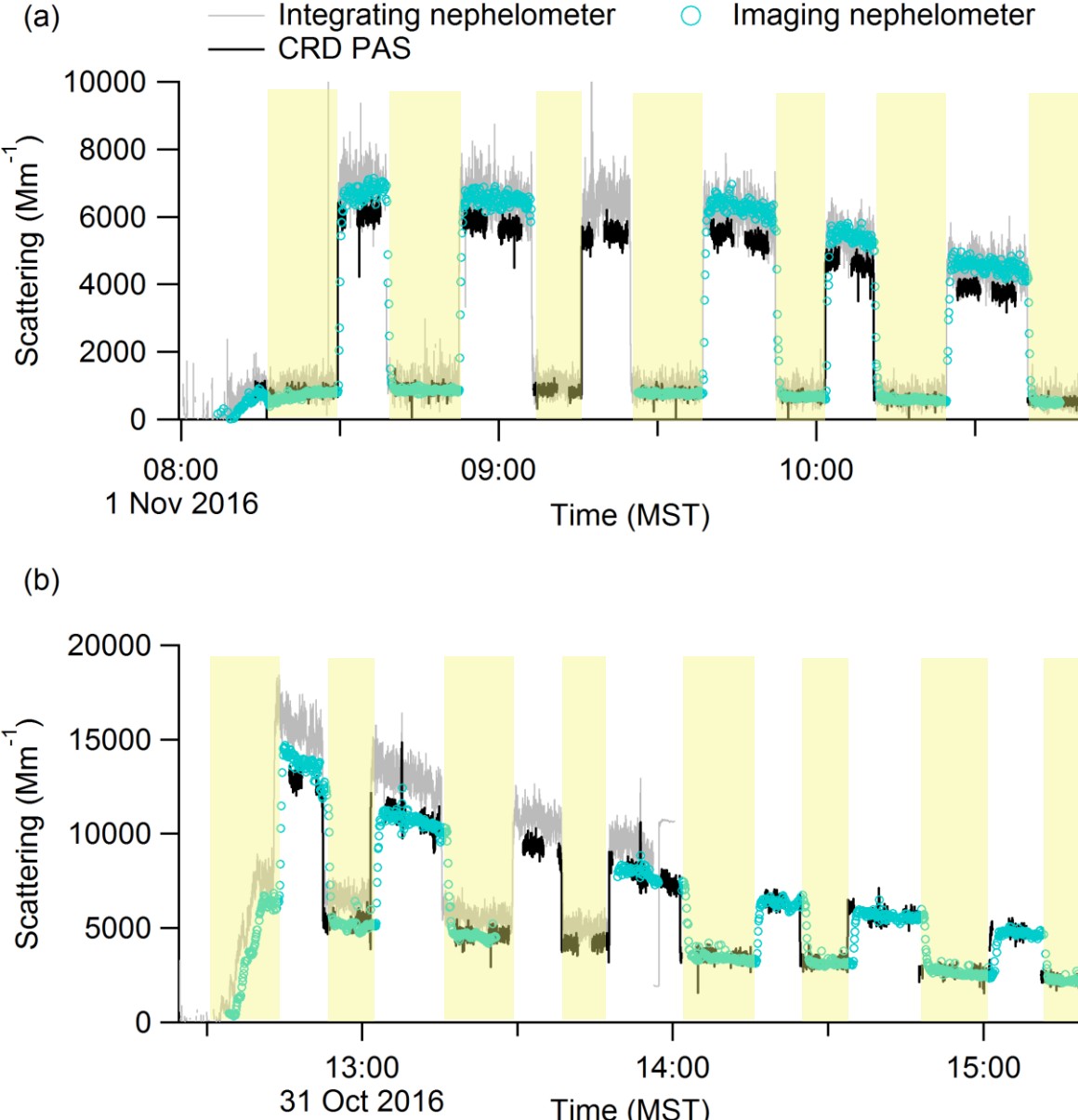

Figure 4: Total integrated scattering for (a) Fire A and (b) Fire B at 405 nm measured by the imaging nephelometer (blue circles), integrating nephelometer (grey line; measured at 450 nm and adjusted to 405 nm), and CRD PAS (black line; extinction 405 nm – absorption 401 nm). Yellow shaded regions indicate when the instruments were sampling behind the thermodenuder. Integrating nephelometer data were unavailable after 13:58 in Fire B (see text).

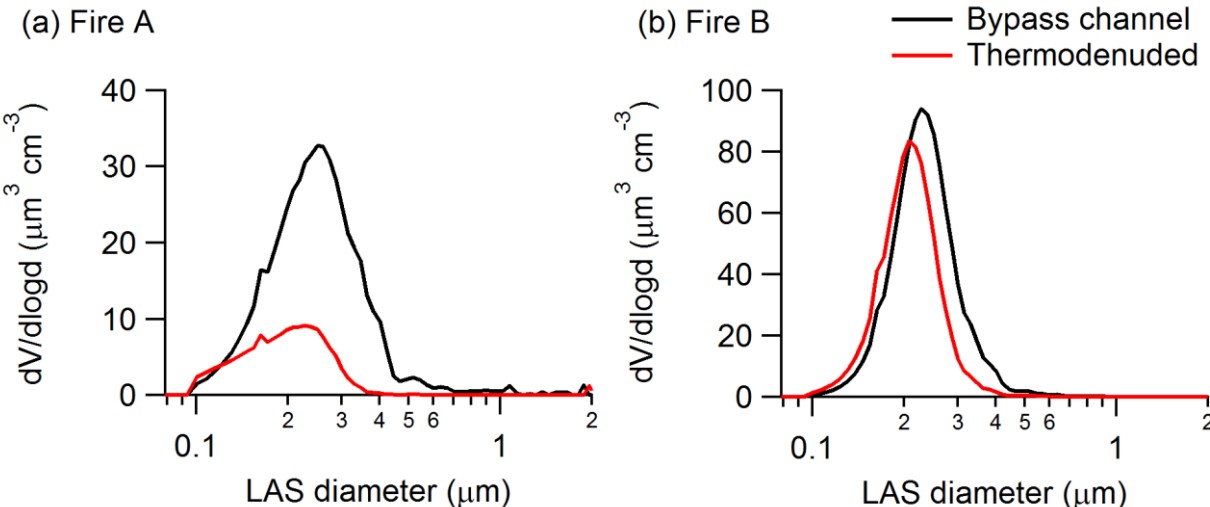

**Figure 5: LAS volume size distribution for (a) Fire A and (b) Fire B showing average over one cycle of bypass channel (black) and after thermodenuder (red).**

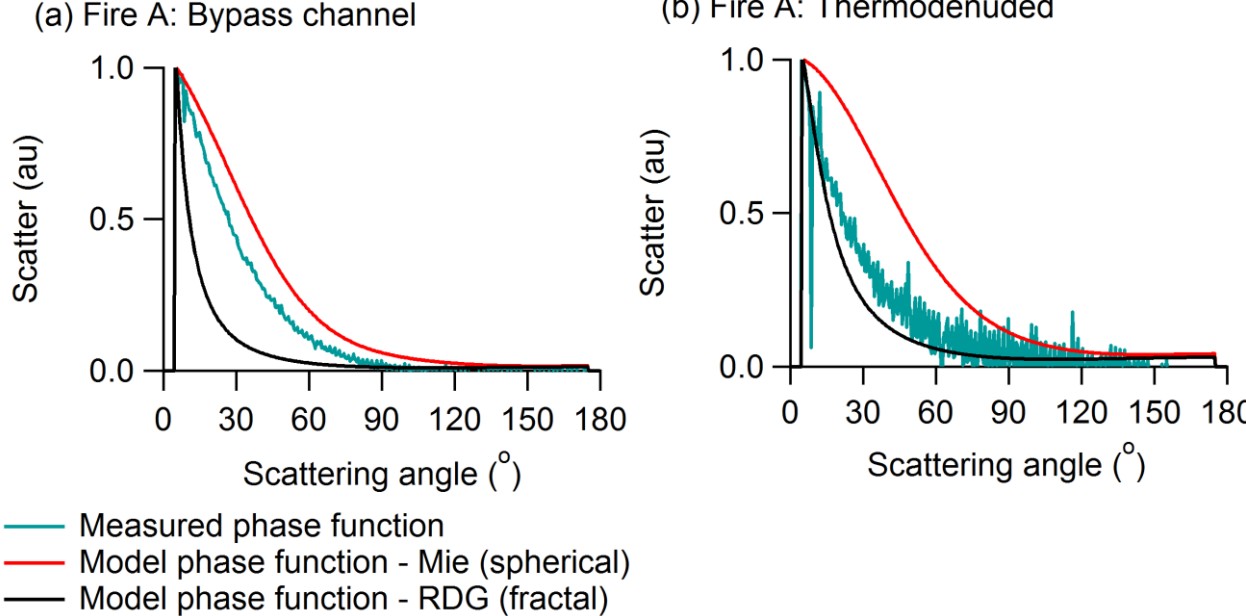

**Figure 6: Comparison of measured (blue) phase function at 405 nm to Mie theory model (red) and RDG model (black) for one cycle of sampling through bypass channel (a) and after the thermodenuder (b) for Fire A. Phase functions are normalized to unity at 5° scattering angle. Mie theory calculations are based on HULIS refractive index (Dinar et al., 2008) and RDG calculations are based on ponderosa pine parameterization (Chakrabarty et al., 2006).**

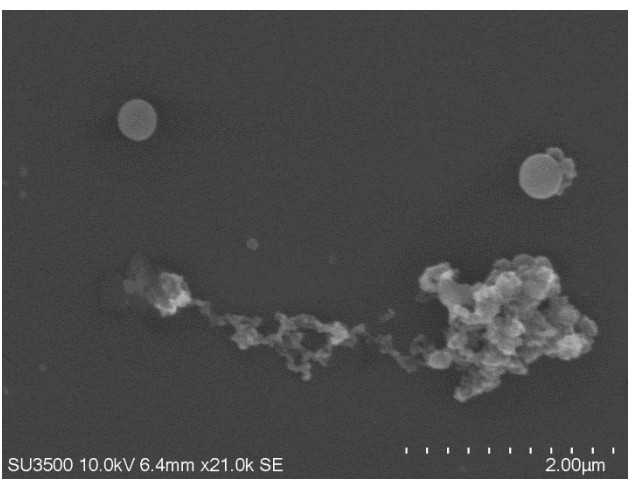

**Figure 7: SEM image of aerosol collected after the thermodenuder during Fire A. While most of the particles appeared fractal-like, some spherical particles (presumably non-volatile organics) were also evident.**

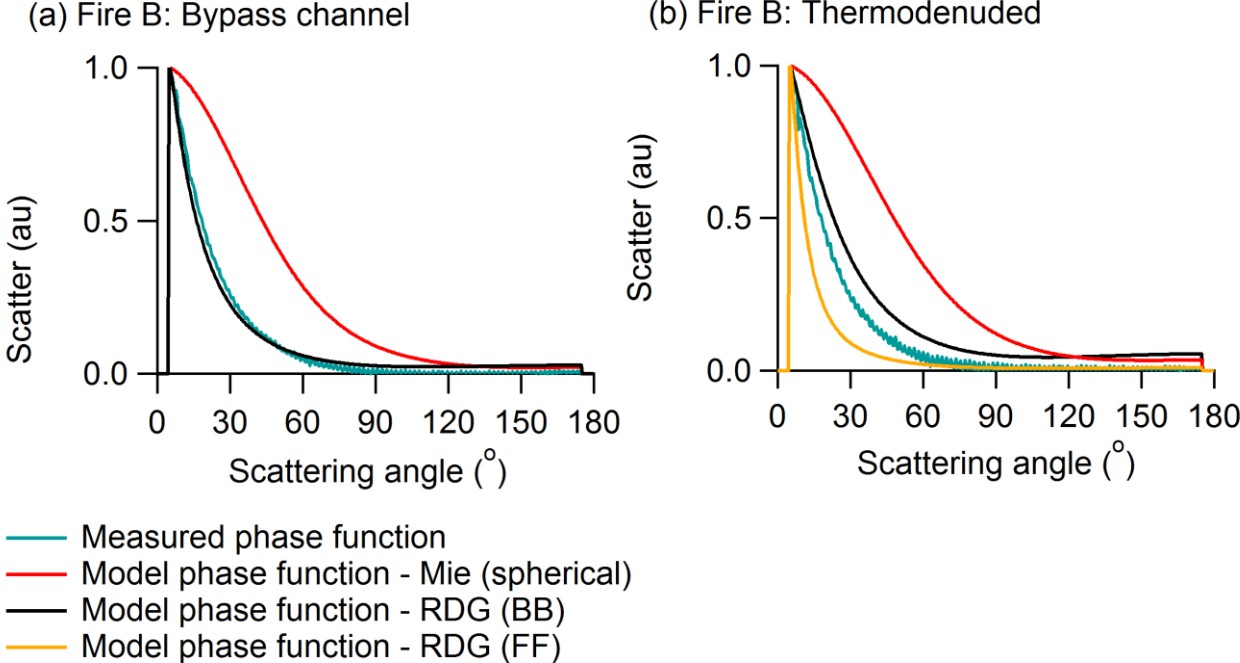

**Figure 8: Comparison of measured (blue) phase function at 405 nm to Mie theory model (red) and RDG models (black and orange) for one cycle of sampling through bypass channel (a) and after the thermodenuder (b) for Fire B. Two RDG parameterizations were compared: fossil fuel (FF; orange) and biomass burning (BB; black). Phase functions are normalized to unity at 5° scattering angle. Mie theory calculations are based on HULIS refractive index (Dinar et al., 2008) and RDG calculations are based on sage fuel for biomass burning (BB) case (Chakrabarty et al., 2006) and laboratory combustion of fossil fuels (FF) (Sorensen, 2001).**