# Peer review of "Investigating biomass burning aerosol morphology using a laser imaging nephelometer"

_Atmospheric Chemistry and Physics, 2017_

## Referee Comment (RC1) · Anonymous Referee #1 · 13 Oct 2017

This paper presents very interesting data on the scattering phase function of aerosol emitted from biomass burning. There is a strong need for a better quantification of the scattering properties of non-spherical particles. I, therefore, fully agree with the authors' sentence on page 12 that "It would be useful to collect more information about how the morphology of fresh emissions evolves with aging in the atmosphere". The data obtained here are very encouraging regarding the ability of this new instrument to provide this kind of data that can aid radiative forcing calculations as well as remote sensing retrievals. I, therefore, strongly support the publication of this paper.

General comments:

The paper is very well written and the experiments have been mostly well described, and for the most part, well justified. I think the paper can be published almost as is,

with just a few minor clarifications and additions that I will discuss next.

Specific comments:

- Introduction

* Page 2, lines 19- 29: An interesting recent approach for determining the shape of coarse particles is provided by Berg et al., Solving the inverse problem for coarse-mode aerosol particle morphology with digital holography. Scientific Reports, 2017. 7(1): p. 9400.

* Page 2 bottom, and 3 top. There are also commercial nephelometers that measure the backscattering at different angles by adjusting the backscattering arm angle. Maybe this could be mentioned here. See, for example, Chamberlain-Ward et al., Advances in Nephelometry through the Ecotech Aurora Nephelometer" TheScientificWorldJOUR-NAL, vol. 11, Article ID 310769, 6 pages, 2011.

- Experimental

* At the beginning of the section: It would be nice to mention the reasons for the choice of the specific wavelengths. Why these and why so close to each other?

* Why is the polarization converted to circular? It would be useful to explain the reason. Could one switch the polarization periodically from vertical to horizontal to collect images at different polarizations and possibly further gain insights on the shape of the particles? If not why? Or is just a practical choice?

* Also, it would be good to provide some details about the beam quality, the noise and the stability of the lasers and provide some information on the set of apertures.

* On page 5 at the top: The Rayleigh scattering contribution is calculated based on pressure and temperature. What composition is assumed here? Is there any potential for biases due to gases specific to the air in which the aerosol are transported? For example, NO2 (although that was probably removed by the scrubber)? In other words,

was there any attempt to compare this with measurements of just aerosol-free air using a filter?

* On page 6, line 27: 5 nm coating is quite thick when compared with ∼50 nm soot monomers (see below on the monomer diameter, as well). That might also affect how coated the particles might appear vs. be "naturally".

* Page 7, line 10: the fractal-like structure depends also on the presence of co-emitted components that might condense on the soot and restructure it.

* Page 7, line 16: It might be good to cite Sorensen's review paper here as well (it is cited a few sentences later on, but it seems to fit here too).

* Page 8, lines 14-17: Some more details here would be useful. How did you get the total scattering from the LAS? How was the match performed? What algorithm was used? etc.

* Page 8, line 17: Monomers diameter might change quite a bit depending on the specific combustion conditions (for example, even for the specific of biomass burning, in the paper by China et al. (2003) cited earlier on in the manuscript, the monomers had diameters from 37 to 56 nm). Did the author estimate the diameter from their SEM images? Would the exact diameter value make a large difference in the comparison when using RDG? Similar comments apply to the effect of k0 and Df. Although some of these aspects are touched upon later in the paper.

- Results and discussion

* At the beginning of the section, while discussing the performance of the instrument: It might be useful, for the purpose of comparing the performance of this instrument to the other more widely used techniques, to also calculate Allan-Werle variances for the TSI and the CRD-PAS instruments.

* Page 9, line 24: a work that might be worth mentioning here is also that by Liu et al. , Aerosol single scattering albedo dependence on biomass combustion efficiency: Laboratory and field studies. Geophysical Research Letters, 2014. 41(2): p. 2013GL058392.

\* Page 10, lines 17-19: Is there any guess on the reasons for the disagreement?

\* Page 10: While discussing figure 4, it would be good to provide some more details (here or earlier on would be fine too) on the details of the RDG calculations.

\* Page 11, lines 4 and 5: "Given the value of the fractal dimension (ðİŘůðİŚŞ = 1.85)..." maybe I misunderstood, but I thought the Df was assumed from other work, not measured somehow in this work, so I am not sure how this conclusion can be reached here.

\* Page 11, line 10: in the paper, cited earlier on in the manuscript, by China et al. 2003, an analysis of the changes of tar balls by the thermal denuder is also discussed.

\* Page 11 at the bottom: a discussion of the effect of thermodenuding on fractal aggregates from different fuels is also provided in a recent paper by Bhandari et al., Effect of Thermodenuding on the Structure of Nascent Flame Soot Aggregates. Atmosphere, 2017. 8(9): p. 166.

- Figure 2. Caption: "The laser axis is curved due to the alignment with respect 5 to the wide-angle lens." Please clarify.

- Figures 6 and 8 How are the different curves normalized?

---

## Referee Comment (RC2) · Anonymous Referee #2 · 21 Oct 2017

This is a very interesting manuscript that describes the performance of a fairly novel polar Nephelometer used of the characterization of fresh biomass burning aerosols and their morphology. This is a generally well-written manuscript but it will need major revisions due to the lack of fractal analysis of SEM images (comments 11, 12) and the selective use of literature values (comments 7, 9, 12). However, it is appropriate for ACP and should be published after the following comments are taken into account.

1. P. 4 LiNeph: As this commercial polar nephelometer has not previously been described in the literature, a more detailed description would be desirable with more information on choice of wavelengths, beam dump, flow rates, curvature of beam image, etc. being of interest.

2. P.4 Image Processing: This section would benefit from quantification and examples

for background spectra, wall scattering, etc.

3. P.5, L.2: replace "aerosol-only" with "particle-only" as "aerosol" is defined as colloid of particles in air or other gases; i.e., aerosol includes the gaseous component.

4. P.4, L.12: "was found to be linear"; please quantify and show an example.

5. P. 7, L.8 and elsewhere "fractal-like particles; and P. 8, L. 4 and elsewhere "fractal aerosols". Please decide if you want to call these particles "fractal-like" or "fractal" and use this consistently. Personally, I prefer "fractal-like" as the agglomerates are not purely fractal; the fractal nature breaks down below a certain length scale (monomer size).

6. P.7,L.18: "and Abs(2ka(m-1)) «1 with wavenumber (k=2pi) and a equal to the monomer radius." Why not just use the definition of the size parameter x for the monomer and replace with "and Abs(x(m-1)) «1 , where x is the monomer size parameter."?

7. P.8, L.4-6: "While fractal aerosols produced from fossil fuel combustion have been studied extensively, only one systematic measurement of the fractal parameterization of fresh biomass burning combustion products has been undertaken. Gwaze et al. (2006) measured k=2.77 and D=1.85 for biomass burning aerosol, although the authors note that 1.85 is a lower limit on Df."

What about the systematic analysis of fractal characteristics of particles emitted from the combustion of different biomass fuels given by Chakrabarty et al. (2006). Please cite and compare with Gwaze et al. (2006).

8. P. 8, L.17: "manufacturers specifications for a similar instrument". Which instrument is this (please specify) and how similar are these instruments? Is the angular range identical?

9. P. 8, L. 21-23: "Spherical particles were assumed to be predominantly organic material, and a representative refractive index for humic-like substances was assumed at

each wavelength (m = 1.64 + 0.12i at 375 nm; m = 1.64 + 0.11i at 405 nm) (Hoffer et al., 2005; Lang-Yona et al., 2009). " Why do you use refractive indices for humic-like substances when refractive indices for OC from biomass burning (with much smaller imaginary parts) are readily available (Chakrabarty et al., 2010) and much more appropriate? Please explain and compare results when using either refractive index.

10. P. 9, L.9: Replace "errors arising from the background signal is more systematic..." with "errors arising from the background signal are more systematic...".

11. P11, L. 6-8: "This implication from the phase function measurements is further supported by SEM images (Figure 7), which show that the non-volatile (denuded) components are mostly fractal-like particles." A quantitative analysis of the SEM images for fractal-like parameters (dimension, pre-factor) is much needed to check applicability of literature values used for optical modeling.

12. P.11, L.17-19: "The phase function measured for the total aerosol population (Figure 8(a)) shows good agreement with the RDG model, assuming the parameterization (k=2.77; D=1.85) from Gwaze et al. (2006) for biomass burning products." This needs to be compared with other literature values (Chakrabarty et al., 2006) and even better with SEM characterization of fractal-like parameters.

13. P.11, L. 32: "the measurements suggest". Specify which measurements.

14. P.25, Fig. 6 and P.27, Fig. 8: The y-axis is labeled with "Scatter (au)", however, the figure caption states "Phase functions are normalized to total scatter in spherical coordinates." Which one is true? Also what do you mean by "scatter"; I'm not familiar with this term (differential scattering cross-section?), please change or explain and give reference.

References

Chakrabarty, R. K., H. Moosmuller, M. A. Garro, W. P. Arnott, J. W. Walker, R. A. Susott, R. E. Babbitt, C. E. Wold, E. N. Lincoln, and W. M. Hao (2006). Emissions from the

Laboratory Combustion of Wildland Fuels: Particle Morphology and Size. J. Geophys. Res., 111, doi:10.1029/2005JD006659.

Chakrabarty, R. K., H. Moosmuller, L.-W. A. Chen, K. Lewis, W. P. Arnott, C. Mazzoleni, M. K. Dubey, C. E. Wold, W. M. Hao, and S. M. Kreidenweis (2010). Brown Carbon in Tar Balls from Smoldering Biomass Combustion. Atmos. Chem. Phys., 10, 6363-6370.
* * *

---

## Author Comment (AC1) · 20 Nov 2017

Response to Reviewers
Manuscript: ACP-2017-842
Manuscript title: Investigating biomass burning aerosol morphology using a laser imaging nephelometer

The discussion below includes the complete text from the reviewer (bold), along with our responses to the specific comments and the corresponding changes (additions in italics, deletions struck through) made to the revised manuscript. All line numbers refer to the original manuscript.

Response to Reviewer #1 Comments:

**This paper presents very interesting data on the scattering phase function of aerosol emitted from biomass burning. There is a strong need for a better quantification of the scattering properties of non-spherical particles. I, therefore, fully agree with the authors' sentence on page 12 that "It would be useful to collect more information about how the morphology of fresh emissions evolves with aging in the atmosphere". The data obtained here are very encouraging regarding the ability of this new instrument to provide this kind of data that can aid radiative forcing calculations as well as remote sensing retrievals. I, therefore, strongly support the publication of this paper.**

**General comments:**

**The paper is very well written and the experiments have been mostly well described, and for the most part, well justified. I think the paper can be published almost as is, with just a few minor clarifications and additions that I will discuss next.**

We are grateful for the reviewer's very positive comments, and appreciate that they took the time to review the manuscript carefully. We have tried to address their suggestions and concerns to the best of our ability individually below.

**Specific comments:**

**- Introduction**

**\* Page 2, lines 19-29: An interesting recent approach for determining the shape of coarse particles is provided by Berg et al., Solving the inverse problem for coarse-mode aerosol particle morphology with digital holography. Scientific reports, 2017. 7(1): p. 9400.**

We thank the reviewer for bringing our attention to this paper, which indeed shows a very sophisticated method to measure morphology of coarse mode particles in an uninterrupted aerosol flow. We have added the following to the appropriate paragraph:

P.2 L.29: *"Recently, a sophisticated method for determining the morphology of coarse mode aerosol using holographic imaging without interrupting the aerosol flow was demonstrated (Berg et al., 2017), however this method is unfeasible for accumulation mode particles."*

**\* Page 2 bottom, and 3 top. There are also commercial nephelometers that measure the backscattering at different angles by adjusting the backscattering arm angle. Maybe this could be mentioned here. See, for example, Chamberlain-Ward et al., Advances in Nephelometry through the Ecotech Aurora Nephelometer" The Scientific World Journal, vol. 11, Article ID 310769, 6 pages, 2011.**

The reviewer is right to point out that we did not mention nephelometers that allow the user to set the backscatter angular coverage. We have adjusted the text (added text in italics, removed text struck through) as follows:

P.3 L.1-3: "…integrating nephelometers (Anderson et al., 1996). *Some nephelometers also have an option to measure set angular regions by blocking portions of the forward scattered light (Chamberlain-Ward and Sharp, 2011).* While these devices are reliable and robust, the  measurements *over large portions of the phase function* do not provide much information about particle morphology due to the lack of angular specificity."

**- Experimental**

**\* At the beginning of the section: It would be nice to mention the reasons for the choice of the specific wavelengths. Why these and why so close to each other?**

The two wavelengths were chosen with the hope of seeing more evidence of absorption by brown carbon particles in the phase functions measured at 375 nm compared to those measured at 405 nm. Unfortunately, as absorption predominantly affects the backscatter, where the signal is relatively low, we were unable to observe wavelength-dependent differences under these experimental conditions. In the future, we may try to do so by increasing the laser power (which would saturate the CCD at forward scatter angles) to have better signal-to-noise in the backscatter region. To address the question, we have added the following to the first paragraph of the Experimental section:

P.4 L.6: "…120 mW respectively. *These wavelengths were selected under the hypothesis that increased absorption by brown carbon particles at 375 nm compared 405 nm may be observable in the measured phase functions, however we were unable to observe this under the experimental conditions.* The output beams…"

**\* Why is the polarization converted to circular? It would be useful to explain the reason. Could one switch the polarization periodically from vertical to horizontal to collect images at different polarizations and possibly gain further insights on the shape of the particles? If not why? Or just a practical choice?**

The use of circularly polarized light is, essentially, a practical decision since it eliminates the need to switch polarization (effectively halving the acquisition rate). In the future, we may adjust the instrument to have this capability since the depolarization ratio provides additional information, particularly for larger (especially dust) particles. We have altered the text as follows:

P.4 L.8: "…sample chamber. *The circular polarization of the beam allows retrieval of the unpolarized phase function, for a homogeneous sample (Bohren and Huffman, 1983), simplifies the experimental set-up by eliminating the need for a variable waveplate (Dolgos and Martins, 2014), and maximizes the data acquisition rate by only recording the phase function under one polarization condition at each wavelength.* A set of apertures…"

**\* Also, it would be good to provide some details about the beam quality, the noise and the stability of the lasers and provide some information on the set of apertures.**

The reviewer raises a few valid points here, which we will address one-by-one.

We can infer the beam quality by looking at images of phase functions due solely to Rayleigh scattering. Since the number density of gas molecules will be homogeneous throughout the beam volume, we

assume that variations in pixel intensity over a single angle bin are due solely to the intensity profile of the laser beam. To demonstrate that the beam profile is Gaussian, we added an additional figure to the Supplementary (shown below), and added a brief reference to it in the main text. Note that the figure labels in the Supplementary have been changed to follow the order of the main manuscript.

P.4 L.9-10: "The two beams traveled the same beam path inside the chamber, with collimated diameters of approximately 3 mm*, and exhibited Gaussian intensity profiles (see Figure S1).*"

[Figure]

*Figure S1: Measured pixel intensity across the 14° scattering angle bin (black) is shown compared with Gaussian fit (red) for a single CCD image (1 sec integration time) when the imaging nephelometer is filled with filtered air and signal arises solely from Rayleigh scattering. The residual (grey) demonstrates the quality of the fit.*

With regards to the laser power stability, we account for any significant ($\geq$ 1 mW) drifts in laser power by reading the power at the diode head for each individual image acquisition (and then apply a calibration factor to convert diode head power to the actual power that reaches the sample chamber). More minor power drifts, if important for the retrieved information, would likely appear as a turning point in the Allan-Werle deviation plots in the Supplementary, which we do not observe over timescales of ~ 1 hour. Short-term fluctuations in power may contribute to noise in the total integrated scattering retrieved for the imaging nephelometer (which is, however, less significant than the noise of the integrating nephelometer measurements), however these are largely removed for the phase functions presented by averaging multiple images.

More details were added about the apertures:

P.4 L.8-9: "A set of *three* apertures *(4 mm diameter) blocked stray*  light before the beam passed through the aerosol sample."

**\* On page 5 at the top: The Rayleigh scattering contribution is calculated based on pressure and temperature. What composition is assumed here? Is there any potential for biases due to gases specific to air in which the aerosol are transported? For example, NO2 (although that was probably**

**removed by the scrubber)? In other words, was there any attempt to compare this with measurements of just aerosol-free air using a filter.**

We have clarified the Rayleigh scattering calculation (see below) in the manuscript. In general, the sample aerosol and dilution air were denuded (including of $NO_2$) and dried, and therefore no corrections were applied to the literature value for Rayleigh scattering of clean dry air. We have added a comment on the maximum possible effect of $CO_2$ (modern-day compared to 1950s standard air) on the Rayleigh scattering cross-section, which was extremely small (0.03%). In the laboratory, we did also measure the Rayleigh scattering of pure $CO_2$ and obtained the same calibration curve, within the noise of the measurements, when using Rayleigh scattering cross-sections from the literature (same source as identified below).

P.4 L.31-32: "This correction factor was determined daily using Rayleigh scattering from dried and filtered ambient air, *and Rayleigh scattering coefficients for dry air from the literature (Bodhaine, 1979; Penndorf, 1957). Since the air was dried and scrubbed, corrections for additional water vapor or $NO_2$ were not needed; likewise, corrections for higher modern-day levels of $CO_2$ were deemed unnecessary since additional $CO_2$ (compared to "standard air" in the 1950s (Penndorf, 1957)) would result in a maximum error of 0.03% in the Rayleigh scattering cross-section.* The consistency of the calibration…"

**\* On page 6, line 27: 5 nm coating is quite thick when compared with ~50 nm soot monomers (see below on the monomer diameter, as well). That might also affect how coated the particles might appear vs be "naturally".**

We used the minimum recommended coating thickness to prevent charging of samples under the electron beam. While it is true that this might make soot particles appear to be less clearly composed of individual monomers, it should not be sufficient to cause freshly emitted soot (i.e. mostly un-collapsed) from appearing very "branchy" (i.e. having low $D_f$ values). We did take into account the coating when retrieving the monomer size from SEM images. While it was difficult to distinguish different monomers, we could measure the narrowest width of fractal branches for a variety of particles (which generally were between $60 - 65$ nm in the raw image). Therefore, we hypothesized that 50 nm was likely representative of the monomer diameter, and this agreed well with the literature. However, we have adjusted the text slightly to account for the uncertainty associated with sputter coating thickness (5 nm is the maximum thickness based on the sputter coating duration):

P. 6 L.27: "…later sputter coated with *up to* 5 nm of platinum…"

**\* Page 7, line 10: the fractal-like structure depends also on the presence of co-emitted components that might condense on the soot and restructure it.**

This is a good point. We have added a justification for why we assumed no significant collapse of black carbon fractals (due to the low relative humidity conditions).

P.7 L.10-11: "Since fresh (< 4 hrs) emissions were measured in the absence of sunlight, it is unlikely that significant oxidative aging occurred in the smoke chamber. *The smoke chamber relative humidity was less than 40% for all fires measured, so we assume there was no significant restructuring of the black carbon agglomerates by organic or sulfuric acids (Xue et al., 2009; Zhang et al., 2008)."*

**\* Page 7, line 16: It might be good to cite Sorensen's review paper here as well (it is cited a few sentences later on, but it seems to fit here too).**

We have added the citation to Sorensen 2001 (alongside the Chakrabarty and Liu citations, P.7 L.16).

**\* Page 8, lines 14-17: Some more details here would be useful. How did you get the total scattering from the LAS? How was the match performed? What algorithm was used? etc.**

We have added the following section of text to the Supplementary (as well as a reference to it in the main manuscript) to hopefully clarify how the LAS measurements were utilized, specifically with regards to fractal particles:

*"The LAS retrieves aerosol size distributions based on scattered 633 nm light. According to the manufacturer's specifications, the light collecting optics cover the solid angles 90° ± 57°, excluding 90° ± 14.8°. This translates to 33° – 75.2° and 104.8° – 147° of the scattering phase function with a maximum azimuthal (out of the plane of the phase function) angle of 57° at 90° scattering angle. This means that, for any given phase function, we can calculate the fraction of scattered light theoretically collected and measured by the LAS detection system. Multiplying this fraction by the integrated scatter in absolute units gives the total scatter measured by the LAS. Since the LAS is calibrated for nominally spherical ammonium sulfate particles, we use Mie theory to determine the total scattered measured per particle for each size bin (99 bins). Then, using the parameterizations of $k_o$, $D_f$, and a described in Section 3.1, we adjust the magnitude of $N_p$ (the number of monomers per agglomerate) to match the scatter measured by the LAS for each size bin (based on RDG calculations). Thus, we can approximately correlate LAS size bins with fractal sizes."*

**\* Page 8, line 17: Monomers diameter might change quite a bit depending on the specific combustion conditions (for example, even for the specific of biomass burning, in the paper by China et al. (2003) cited earlier on in the manuscript, the monomers had diameters from 37 to 56 nm). Did the author estimate the diameter from their SEM images? Would the exact diameter value make a large difference in the comparison when using RDG? Similar comments apply to the effect of K0 and Df. Although some of these aspects are touched upon later in the paper.**

As mentioned in the manuscript, we did estimate the spherule size from SEM micrographs (taking into account the added Pt coating thickness). While the RDG phase functions are sensitive to $D_f$, $k_o$, and a, we chose not to comprehensively cover the full range of possible values for the RDG parameterization since the aim of the paper is mainly to serve as a demonstration of the instrumental capabilities. We have added a section to the text to try to convey this more clearly:

P.8 L.19-20: "As mentioned, the values of $D_f$ and $k_o$ were taken to be 1.85 and 2.77, respectively, based on reported fractal-like aerosol measurements in the literature, particularly for biomass burning emissions. *While the phase functions modeled using RDG are sensitive to these values, as well as the monomer diameter, a comprehensive analysis of the full range of possible RDG parameterization is beyond the scope of this paper; rather, here we present an initial "best guess" of the RDG parameterization based on the limited literature on biomass burning-produced soot and SEM micrographs.* Spherical particles were assumed to be predominantly organic material…"

**- Results and discussion**

**\* At the beginning of the section, while discussing the performance of the instrument: It might be useful, for the purpose of comparing the performance of this instrument to the other more widely used techniques, to also calculate Allan-Werle variances for the TSI and the CRD-PAS instruments.**

To better convey the relative performance of the three scattering measurements, we have included the following text and figure in the Supplementary:

"Figure  *S4* shows an Allan-Werle deviation plot for scattering signal integrated over a single angle bin (each ~0.5°) at several measurement angles. This was determined by continuously imaging the phase function of a clean air sample and removing the Raleigh scattering component to measure a "zero" phase function. *Figure S5 shows the Allan-Werle deviation plot for the total integrated scatter at 405 nm for the TSI integrating nephelometer (red) and laser imaging nephelometer (black). Note that the integrating nephelometer has an acquisition rate of 1 Hz for each channel, while the imaging nephelometer has an average acquisition rate of 0.2 Hz. The integrating nephelometer scattering was scaled from 450 nm (observed) to 405 nm using the measured scattering Ångstrom exponent between 450 and 550 nm. The normal mode of operation for the CRD PAS during these experiments included automatic re-zeroing every six minutes, therefore these data are not included in the Allan-Werle plot; the accuracy of both the absorption and extinction channels is ±5%. Details of the CRD PAS can be found in Lack et al. (2012)."*

[Figure]

*Figure S5: Allan-Werle deviation plot of integrated scatter at 405 nm for the imaging nephelometer (black) and integrating nephelometer (red). The integrating nephelometer measurements at 450 nm were scaled to 405 nm using the measured scattering Ångstrom exponent between 450 and 550 nm. Instruments were co-sampling filtered air.*

**\* Page 9, line 24: a work that might be worth mentioning here is also that by Liu et al., Aerosol single scattering albedo dependence on biomass combustion efficiency: Laboratory and field studies. Geophysical Research Letters, 2014. 41(2): p.2013GL058392.**

We thank the reviewer for the suggestion, the Liu et al. citation and a reference to Pokhrel et al. 2016 have been added (along with Selimovic et al. 2017, P.9 L.24).

**\* Page 10, lines 17-19: Is there any guess on the reasons for the disagreement?**

We investigated the reason for the disagreement and found that it arose from the image processing. We have improved the image processing algorithm (which mainly involved changing the order in which corrections were applied). We have adjusted the text in Sections 2.2 and 4.2 accordingly:

Section 2.2:

P.4 L.1 – P.5 L.3: "The raw CCD images were processed in five steps to produce an aerosol scattering phase function by correcting for the CCD dark background, scattered light due to internal surfaces, and Rayleigh scattering.  First, a dark background spectrum was subtracted from each CCD image.  This is necessary because CCD detectors produce non-zero dark current in the absence of light due to thermal noise and an electronic offset.  The dark background image was acquired with the same integration time as the aerosol images and was measured before each fire.  Second, light scattering from surfaces within the sample volume was removed by subtracting a laser power-normalized correction from each image. The scattering correction was determined by filling the chamber with helium, which has a negligible scattering coefficient, and taking an average of ~20 images at each wavelength in order to account for light reaching the CCD from the instrument body and optics. Third, *each strip of the corrected image was integrated and an additional constant offset likely due mainly to temperature variations in the CCD pixel array was removed. Fourth, the Rayleigh scatter contribution of the dry air, which was measured before each experiment, was pressure-corrected and removed from the signal to determine the aerosol-only scattering.*  *Finally,*  an angle-dependent correction factor *was* applied. This correction factor was determined daily using Rayleigh scattering from dried and filtered ambient air. The consistency of the calibration factor from day-to-day (<3% variation) implies that the lens, window, and chamber walls did not accumulate particles or become dirty over time.  "

Section 4.2:

P.9 L.32 – P.10 L.2: " *The* good agreement between *all three measurements*  provide*s* confidence in the total scatter retrieved by the imaging nephelometer (see Figure  *S6* in Supplementary for correlation plot)."

P.10 L.17-19: " *The CRD PAS and imaging nephelometer agree well throughout this experiment. The higher scattering retrieved by the integrating nephelometer is likely due to errors arising from extrapolating from 450 nm.*"

[Figure]

**\* Page 10: While discussing figure 4, it would be good to provide some more details (here or earlier on would be fine too) on the details of the RDG calculations.**

We have added in references that the reader can look to for more information on the formulae involved in calculating the scattering phase function for the RDG model in Section 3.1:

P.8 L.8-9: "The scattering angle-dependent structure factor, *which is needed to calculate the*  RDG phase function, was adopted from Sorensen et al. (Liu et al., 2013a; Sorensen et al., 1992; Yang and Köylü, 2005). *Details of the phase function calculations based on the RDG model can be found in Liu et al. (2013a) and Kandilian et al. (2015)."*

**\* Page 11, lines 4 and 5: "Given the value of the fractal dimension ($D_f$ = 1.85)…" maybe I misunderstood, but I thought the Df was assumed from other work, not measured somehow in this work, so I am not sure how this conclusion can be reached here.**

We apologize for the confusion, and realize that without performing a thorough sensitivity analysis it is a bit over-stretching to make statements about fractal collapse based on the qualitative comparison between the RDG model and measurement. We have removed the following sentence:

P.11 L.4-6: "~~Given the value of the fractal dimension ($D_f = 1.85$) is close to the minimum diffusion-limited case (~1.75), we assume the agglomeration was (nearly) diffusion-limited, and no significant restructuring or collapse of the particles occurred in the chamber.~~"

**\* Page 11, line 10: in the paper, cited earlier on in the manuscript, by China et al. 2003, an analysis of the changes of tar balls by the thermal denuder is also discussed.**

The China et al. citation has been added (P.11 L.10-11).

**\* Page 11 at the bottom: a discussion of the effect of thermodenuding on fractal aggregates from different fuels is also provided in a recent paper by Bhandari et al., Effect of Thermodenuding on the Structure of Nascent Flame Soot Aggregates. Atmosphere, 2017. 8(9): p. 166.**

We are grateful for the very helpful suggestion to look at this paper. We have added the following text:

P. 11 L.30-31: "…(Skillas et al., 1998). *Additionally, soot agglomerates emitted by ethylene and methane flames did not exhibit restructuring after being heated to 250 – 270°C (Bhandari et al., 2017).* If the fractal particles underwent restructuring.."

**- Figure 2. Caption: "The laser axis is curved due to the alignment with respect to the wide-angle lens." Please clarify.**

We have tried to clarify this caption as follows:

P.21 L.5: " *The blue dashed line shows the center of the laser beam; the slight curvature arises from the wide-angle lens being slightly off-center relative to the laser path.*"

**- Figures 6 and 8 How are the different curves normalized?**

In the literature, phase functions are typically normalized either for the total integral over all solid angles equal to unity or forward scatter (0°) equal to unity. We understand that the former method (which was used in the original manuscript) may be somewhat confusing, and is also more sensitive to noise in the side-scattering region where the signal-to-noise-ratio is relatively small. Therefore, in the updated draft we have selected the second method of normalization (normalizing to 5° scattering angle because we cannot measure at 0°). The new figures (P.25,27), with updated captions, are shown here:

[Figure]

**Figure 1: Comparison of measured (blue) phase function at 405 nm to Mie theory model (red) and RDG model (black) for one cycle of sampling through bypass channel (a) and after the thermodenuder (b) for Fire A. ** *Phase functions are normalized to unity at 5° scattering angle. Mie theory calculations are based on HULIS refractive index (Dinar et al., 2008) and RDG calculations are based on ponderosa pine parameterization (Chakrabarty et al., 2006).*

[Figure]

**Figure 2: Comparison of measured (blue) phase function at 405 nm to Mie theory model (red) and RDG models (black and orange) for one cycle of sampling through bypass channel (a) and after the thermodenuder (b) for Fire B. Two RDG parameterizations were compared: fossil fuel (FF; orange) and biomass burning (BB; black). ** *Phase functions are normalized to unity at 5° scattering angle. Mie theory calculations are based on HULIS refractive index (Dinar et al., 2008) and RDG calculations are based on sage fuel for biomass burning (BB) case (Chakrabarty et al., 2006) and laboratory combustion of fossil fuels (FF) (Sorensen, 2001).*

---

## Author Comment (AC2) · 20 Nov 2017

Response to Reviewers
Manuscript: ACP-2017-842
Manuscript title: Investigating biomass burning aerosol morphology using a laser imaging nephelometer

The discussion below includes the complete text from the reviewer (bold), along with our responses to the specific comments and the corresponding changes (additions in italics, deletions struck through) made to the revised manuscript. All line numbers refer to the original manuscript.

Response to Reviewer #2 Comments:

**This is a very interesting manuscript that describes the performance of a fairly novel polar Nephelometer used of the characterization of fresh biomass burning aerosols and their morphology. This is a generally well-written manuscript but it will need major revisions due to the lack of fractal analysis of SEM images (comments 11, 12) and the selective use of literature values (comments 7, 9, 12). However, it is appropriate for ACP and should be published after the following comments are taken into account.**

We are grateful for the reviewer's careful consideration of the manuscript and appreciate their feedback. We have attempted to address his/her concerns and make the recommended revisions, as seen below.

**1. P. 4 LiNeph: As this commercial polar nephelometer has not previously been described in the literature, a more detailed description would be desirable with more information on choice of wavelengths, beam dump, flow rates, curvature of beam image, etc. being of interest**

The two wavelengths were chosen with the hope of seeing more evidence of absorption by brown carbon particles in the phase functions measured at 375 nm compared to those measured at 405 nm. Unfortunately, as absorption predominantly affects the backscatter, where the signal is relatively low, we were unable to observe these effects under these experimental conditions. In the future, we may try to do so by increasing the laser power (which would saturate the CCD at forward scatter angles) to have better signal-to-noise in the backscatter region. To address the question, we have added the following to the first paragraph of the Experimental section:

P.4 L.6: "…120 mW respectively. *These wavelengths were selected under the hypothesis that increased absorption by brown carbon particles at 375 nm compared 405 nm may be observable in the measured phase functions, however we were unable to observe this under the experimental conditions.* The output beams…"

We clarify the curvature of the beam in the Figure 2 caption as follows:

P.21 L.5: " *The blue dashed line shows the center of the laser beam; the slight curvature arises from the wide-angle lens being slightly off-center relative to the laser path.*"

The flow rates are given in Section 2.3, since this is somewhat dependent on the type of experiment.

**2. P. 4 Image Processing: This section would benefit from quantification and examples for background spectra, wall scattering, etc.**

We deepened discussion of the image processing approach with the following text and figure in the Supplementary. Note that the Supplementary figures have been renumbered to follow the sections of the main manuscript.

*"Figure S2 depicts the relative contribution of different signals to the raw measurements. The data have all been summed across each angle bin. The raw data (black line) represent a single measurement of the total aerosol population during Fire A (see Section 4.1). The non-particle contributions are shown in colors and stacked. The Rayleigh scattering and helium correction have been scaled for laser power during the measurement, and the Rayleigh scattering has also been corrected for pressure. We note that a much higher laser power is used when measuring only Rayleigh scattering for calibration measurements. The dark background is predominantly accounted for by the voltage across the CCD; the standard deviation across the pixel array and variation for different acquisition times (100 ms compared to 10 s) accounts for < 5% of this signal level. The dark background appears to contribute substantially to the signal here because we have summed the signal across all pixels, including those not exposed to scattering from the laser beam (see Figure 2 in the main manuscript)."*

[Figure]

*Figure S2: Contribution of various sources of signal to a single measurement of scattering of the total aerosol population (bypass channel) during Fire A. The black line indicates the raw signal summed across all pixels for each angle bin. The coloured lines show the corrections applied, and are stacked in the following order: dark background (grey), background scatter (likely caused by temperature variations of the CCD and multiple scatter from the walls), helium correction (direct wall scatter), and Rayleigh contribution. The dark background is dominated by the signal associated with the voltage across the CCD; read-out and single-shot thermal noise together account for <5%. The Rayleigh and helium corrections have been corrected for laser power, and the Rayleigh contribution has also been corrected for pressure. The integrated scatter from the aerosol particles was measured to be 88.3 Mm$^{-1}$ in the instrument corresponding to 4741 Mm$^{-1}$ in the original sample after accounting for dilution.*

**3. P.5, L.2: replace "aerosol-only" with "particle-only" as "aerosol" is defined as colloid of particles in air or other gases; i.e., aerosol includes the gaseous component.**

Done.

**4. P.4, L.12: "was found to be linear"; please quantify and show an example.**

The following image and text have been added to the Supplementary and referenced in the main manuscript:

*"Figure S3 shows a sample pixel-to-angle calibration curve for the 405 nm laser determined by comparing the pixel associated with the measured maxima and minima with the angles of the phase function maxima and minima predicted by Mie theory for monodisperse PSLs with diameters of 520, 600, and 700 nm."*

[Figure]

*Figure S3: Pixel-to-angle calibration plot for 405 nm showing angles (calculated using Mie theory) versus measured indices for maxima and minima appearing in phase functions for monodisperse PSL: 520 nm (green), 600nm (orange), and 700 nm (red). The fit, including all sizes, has a slope of 0.484, intercept of -5.45, and $R^2$ value of 0.997.*

**5. P. 7, L.8 and elsewhere "fractal-like particles; and P. 8, L. 4 and elsewhere "fractal aerosols". Please decide if you want to call these particles "fractal-like" or "fractal" and use this consistently. Personally, I prefer "fractal-like" as the agglomerates are not purely fractal; the fractal nature breaks down below a certain length scale (monomer size).**

We thank the reviewer for the suggestions. We have harmonized the text to use "fractal-like" when discussing particles and "fractal" when referring specifically to theoretical models.

**6. P.7,L. 18: "and Abs(2ka(m-1)) << 1 with wavenumber (k=2pi) and a equal to the monomer radius." Why not just use the definition of the size parameter x for the monomer and replace with "and Abs (x(m-1)) << 1, where x is the monomer size parameter."?**

We have made the following changes:

P.7 L.18: "… and $|2ka(m-1)| \ll 1$  $|2x_P(m-1)| \ll 1$ *with $x_P$ equal to the size parameter of the monomer.*"

**7. P.8, L.4-6: "While fractal aerosols produced from fossil fuel combustion have been studied extensively, only one systematic measurement of the fractal parameterization of fresh biomass burning combustion products has been undertaken. Gwaze et al. (2006) measured k=2.77 and D=1.85 for biomass burning aerosol, although the authors note that 1.85 is a lower limit on Df."**

**What about the systematic analysis of fractal characteristics of particles emitted from the combustion of different biomass fuels given by Chakrabarty et al. (2006). Please cite and compare with Gwaze et al. (2006).**

We thank the reviewer for pointing out this paper. We have incorporated this study into our discussion in Sections 3.1 and 4.3.

P.8 L.4-23: "While fractal aerosols produced from fossil fuel combustion have been studied extensively,  *few* systematic measurement*s* of the fractal parameterization of fresh biomass burning combustion products *have*  been undertaken. *Chakrabarty et al. (2006) reported $k_o$ values from 2.05 – 2.90 and $D_f$ values in the range of 1.67 – 1.83 for various biomass fuels.* Gwaze et al. (2006) measured $k_o = 2.77$ and $D_f = 1.85$ for biomass burning aerosol, although the authors note that 1.85 is a lower limit on $D_f$.  *In this work, unless otherwise noted, $k_o$ and $D_f$ are taken from Chakrabarty et al. (2006) based on fuel type.* The scattering angle-dependent structure factor, which allows calculating a RDG phase function, was adopted from Sorensen et al. (Liu et al., 2013a; Sorensen et al., 1992; Yang and Köylü, 2005). *Details of the phase function calculations based on the RDG model can be found in Liu et al. (2013a) and Kandilian et al. (2015).*

We used measured size distributions … For RDG model calculations, the monomer diameter ($2a$) was assumed to be 50 nm, based on SEM images and in agreement with *past studies of biomass burning (Chakrabarty et al., 2006; Gwaze et al., 2006)* . A wavelength-independent refractive index for black carbon ($m = 1.95 + 0.8i$) was assumed (Bond and Bergstrom, 2006).  *The values of $D_f$ and $k_o$ were taken from the literature for similar fuel types and are detailed below (Chakrabarty et al., 2006). While the phase functions modeled using RDG are sensitive to these values, as well as the monomer diameter, a comprehensive analysis of the full range of possible RDG parameterization is beyond the scope of this paper; rather, here we present an initial "best guess" of the RDG parameterization based on the limited literature on biomass burning-produced soot and SEM micrographs.* Spherical particles were assumed to be predominantly organic material, and a representative refractive index for humic-like substances was assumed at each wavelength ($m = 1.64 + 0.12i$ at 375 nm; $m = 1.64 + 0.11i$ at 405 nm) (Hoffer et al., 2005; Lang-Yona et al., 2009). *A comparison of Mie theory calculations for a range of refractive indices is available in the Supplementary.*"

P.10 L.29-P.11 L.4: "Figure 6 shows measured phase functions … readily form spherical particles and be well-represented by Mie theory. *A comparison of phase functions assuming different refractive indices and fractal parameterizations is available in the Supplementary.* In contrast, the phase function of the denuded products looks much more similar to the RDG fractal model prediction based on the *ponderosa pine parameterization ($k_o = 2.32; D_f = 1.69$) from Chakrabarty et al. (2006)*  (Figure 6(b)). This would suggest that the refractory material has a predominantly fractal-like morphology that is well described using this parameterization for fresh biomass burning emissions."

P.11 L.17-19, L.33: "… the total aerosol population (Figure 8(a)) shows good agreement with the RDG model, assuming the *parameterization for sage fuel ($k_o = 2.56; D_f = 1.79$) from Chakrabarty et al. (2006).*  The agreement is

less good for … indicating no significant restructuring in the thermodenuder. *Unfortunately, no samples from Fire B were collected for SEM analysis.*"

**8. P. 8, L. 17: "manufacturers specifications for a similar instrument". Which instrument is this (please specify) and how similar are these instruments? Is the angular range identical?**

Since submitting the original manuscript, we found the manufacturer's specifications for the LAS (the instrument used here), and so have noted that the angular range comes directly from the specifications for this instrument.

**9. P. 8, L. 21-23: "Spherical particles were assumed to be predominantly organic material, and a representative refractive index for humic-like substances was assumed at each wavelength (m = 1.64 + 0.12i at 375 nm; m = 1.64 + 0.11i at 405 nm) (Hoffer et al., 2005; Lang-Yona et al., 2009). " Why do you use refractive indices for humic-like substances when refractive indices for OC from biomass burning (with much smaller imaginary parts) are readily available (Chakrabarty et al., 2010) and much more appropriate? Please explain and compare results when using either refractive index.**

We thank the reviewer for the comment. We agree with the reviewer that the HULIS refractive index values represent a relatively absorbing OC, compared to the some of the brown carbon (BrC) studies. Nonetheless, organic carbon (OC) from biomass burning does not have a clear definition, nor a refractive index that is agreed upon across the literature. OC could include both tar balls (defined operationally as a type of near-spherical atmospheric aerosol particle consisting of amorphous carbonaceous material, which have been found to exist in abundance in polluted continental air masses) and BrC. Previous studies have found a wide range of possible imaginary refractive indices for BrC (Kirchstetter et al., 2004; Lack et al., 2012; Alexander et al., 2008; Saleh et al., 2014). For example Kirchstetter et al. (2004) reports highly absorbing k values at 400 nm (0.112) for organic carbon from biomass smoke, whereas Lack et al. (2012) shows very low k values at 404 nm for biomass burning (0.007 ± 0.005). The refractive index for OC given in Chakrabarty et al. (2010) is for what they term "tar balls". Chakrabarty et al. (2010) indeed show that tarballs are relatively non-absorbing (m = 1.78 + 0.015i and m = 1.83 + 0.0076i at 405 nm for two types of duff fuels). However, the definition of tar balls is vague, and their chemical composition and degree of volatility vary across the literature. Other studies found tar balls from biomass burning, to be highly absorbing (e.g. Alexander et al. (2008) reported a refractive index of 1.67 – 0.27i at 550 nm). The absorption, as well as the definition, of tar balls in the literature is still not clear.

We believe that the HULIS refractive indices represent intermediate values for these type of aerosols. In an effort towards completeness, we have added a comparison of phase functions for different refractive indices (Mie theory) and fractal parameterizations (RDG) to the Supplementary, and references to the figure in Sections 3.1 and 4.3. This figure includes the two refractive indices given for tar balls produced by duff combustion in the Chakrabarty et al. (2010) paper, as well as a completely non-absorbing aerosol as a reference. It should be evident from this graph that the refractive index of the organic material does not strongly influence the shape of the phase function predicted for spherical particles.

*"Figure S7 shows a comparison of phase functions modeled using two different fractal parameterizations (RDG model) and four different refractive indices for spherical particles (Mie theory). This plot corresponds to Figure 6(a) in the main manuscript. It is evident that the values used do not significantly alter the general shape of the phase functions, with the RDG model always predicting much more strongly forward-scattering phase functions. Details of the parameterizations, including references, are noted in the caption."*

[Figure]

*Figure S7: Intercomparison of measured and modelled phase functions using different parameterizations for Fire A (bypass channel) at 405 nm: measured (red); Mie theory using refractive index for ammonium sulfate (light blue) (Haynes, 2013), ponderosa pine duff (PPDuff; light green) (Chakrabarty et al., 2010), Alaskan duff (AKDuff; dark green) (Chakrabarty et al., 2010), and humic-like substances (dark blue) (Dinar et al., 2008); RDG theory using fractal parameterization for ponderosa pine (grey) (Chakrabarty et al., 2006) and beech (black) (Gwaze et al., 2006). All curves were normalized to unity at 5° scattering angle.*

**10. P.9, L.9: Replace "errors arising from the background signal is more systematic…" with "errors arising from the background signal are more systematic…".**

The sentence has been fixed (P.9 L.9).

**11. P11, L. 6-8: "This implication from the phase function measurements is further supported by SEM images (Figure 7), which show that the non-volatile (denuded) components are mostly fractal-like particles." A quantitative analysis of the SEM images for fractal-like parameters (dimension, pre-factor) is much needed to check applicability of literature values used for optical modeling.**

While we would ideally like to incorporate a thorough quantitative fractal analysis to support our RDG parameterizations, unfortunately we are not able to do this for three reasons: (1) the samples were sputter coated with ~2 – 5 nm of Pt to prevent charging under the strong electron beam, preventing an accurate assessment of monomer overlap; (2) we did not perform a comprehensive study on whether the deposition onto the silicon substrate caused any deformation of the agglomerates or how many contact points the agglomerates have with the substrate; and (3) there is evidence of agglomeration post-deposition (agglomerates >> impactor cut point) concentrated at the center of the impaction site making it difficult to accurately assess the original fractal properties. However, we appreciate that some fractal analysis is beneficial to supporting our parameterization of the RDG model. We have therefore modified the following discussion in Section 4.3:

P.11 L.4-11: "~~Given the value of the fractal dimension ($D_f$ = 1.85) is close to the minimum diffusion-limited case (~1.75), we assume the agglomeration was (nearly) diffusion-limited, and no significant restructuring or collapse of the particles occurred in the chamber.~~ This implication from the phase function measurements is further supported by SEM images (Figure 7), which show that the non-volatile

(denuded) components are mostly fractal-like particles. *Using the SEM software, the average monomer diameter was measured for monomers clearly visible on the outer edges of the agglomerates; the average monomer diameter (excluding Pt coating) is 50 ± 10 nm. A simple box-counting fractal analysis was performed using image analysis software (ImageJ, (Image Processing and Analysis in Java), National Institutes of Health) on the fractal-like particles measured from Fire A (thermodended) to estimate the fractal dimension (Karperien, 2013). This algorithm calculates a fractal dimension ($D_{fB}$) based on the relationship between the length scale (ε) and number of boxes containing a portion of the fractal at each scale ($N_ε$) (Theiler, 1990):*

$$D_{fB} \approx \frac{log(N_\epsilon)}{log(\epsilon)}. \hspace{3cm} (2)$$

*The fractal dimension retrieved (1.87 ± 0.06) likely represents an upper limit as the Pt coating on the particles will increase apparent overlap of monomers. Both the monomer size and fractal dimension (considering it is an upper limit) are in line with the literature values used for the RDG models based on previous studies of biomass burning particles (Chakrabarty et al., 2006; Gwaze et al., 2006). Due to the coating, we are not able to retrieve the fractal dimension and prefactor using other methods previously demonstrated for fractal-like aerosol particles (Brasil et al., 1999; Chakrabarty et al., 2006). Additionally, it is unclear whether any deformation of the fractal-like agglomerates occurred upon deposition to the silicon substrate. Interestingly,* there also appear to be a small fraction of spherical particles in the SEM micrographs that could fall into the classification of "tar balls" – amorphous, non-volatile organic aerosol previously observed in biomass burning aerosol populations that survive thermal denuding (Adachi and Buseck, 2011; Chakrabarty et al., 2010; *China et al., 2013;* Hand et al., 2005; Pósfai et al., 2004)."

**12. P.11, L.17-19: "The phase function measured for the total aerosol population (Figure 8(a)) shows good agreement with the RDG model, assuming the parameterization (k=2.77; D=1.85) from Gwaze et al. (2006) for biomass burning products." This needs to be compared with other literature values (Chakrabarty et al., 2006) and even better with SEM characterization of fractal-like parameters.**

Please refer to Comments 7 and 11 above.

**13. P.11, L. 32: "the measurements suggest". Specify which measurements.**

We clarify by making the following changes:

P.11 L.32-33: "However, *the shape of the measured phase function is qualitatively in agreement with models assuming fractal dimensions of 1.75 – 1.79 (Figure 8(b)), suggesting*  that the fractal dimension does not change *and, consequently,*  no significant restructuring in the thermodenuder."

**14. P.25, Fig. 6 and P.27, Fig. 8: The y-axis is labeled with "Scatter (au)", however the figure caption states "Phase functions are normalized to total scatter in spherical coordinates." Which one is true? Also what do you mean by "scatter"; I'm not familiar with this term (differential scattering cross-section?), please change or explain and give reference.**

In the literature, phase functions are typically normalized either for the total integral over all solid angles equal to unity or forward scatter equal to unity. We understand that the former method (which was used in the original manuscript) may be somewhat confusing, and is also more sensitive to noise in the side-scattering region where the signal-to-noise-ratio is relatively small. Therefore, in the updated draft we have selected the second method of normalization. As is commonly done, we have used arbitrary units

(au) on the axis, however "scatter" informs the reader that the normalized signal is proportional to the total scatter (in units of, i.e. Mm⁻¹). The new figures, with updated captions, are shown here:

[Figure]

**Figure 1: Comparison of measured (blue) phase function at 405 nm to Mie theory model (red) and RDG model (black) for one cycle of sampling through bypass channel (a) and after the thermodenuder (b) for Fire A.** *Phase functions are normalized to unity at 5° scattering angle. Mie theory calculations are based on HULIS refractive index (Dinar et al., 2008) and RDG calculations are based on ponderosa pine parameterization (Chakrabarty et al., 2006).*

[Figure]

**Figure 2: Comparison of measured (blue) phase function at 405 nm to Mie theory model (red) and RDG models (black and orange) for one cycle of sampling through bypass channel (a) and after the thermodenuder (b) for Fire B. Two RDG parameterizations were compared: fossil fuel (FF; orange) and biomass burning (BB; black).** *Phase functions are normalized to unity at 5° scattering angle. Mie theory calculations are based on HULIS refractive index (Dinar et al., 2008) and RDG calculations are based on sage fuel for biomass burning (BB) case (Chakrabarty et al., 2006) and laboratory combustion of fossil fuels (FF) (Sorensen, 2001).*

**References**

Chakrabarty, R. K., H. Moosmuller, M. A. Garro, W. P. Arnott, J. W. Walker, R. A. Susott, R. E. Babbitt, C. E. Wold, E. N. Lincoln, and W. M. Hao (2006). Emissions from the Laboratory Combustion of Wildland Fuels: Particle Morphology and Size. J. Geophys. Res. 111, doi:10.1029/2005JD006659.

Chakrabarty, R. K., H. Moosmuller, L.-W. A. Chen, K. Lewis, W. P. Arnott, C. Mazzoleni, M. K. Dubey, C. E. Wold, W. M. Hao, and S. M. Kreidenweis (2010). Brown Carbon in Tar Balls from Smoldering Biomass Combustion. Atmos. Chem. Phys., 10, 6363-6370.

Alexander, D. T. L., P. A. Crozier, and J. R. Anderson (2008). Brown carbon spheres in East Asian outflow and their optical properties. Science, 321, 833-836, doi:10.1126/science.1155296.

Saleh, R., M. Marks, J. Heo, P. J. Adams, N. M. Donahue, and A. L. Robinson (2014). Contribution of brown carbon and lensing to the direct radiative effect of carbonaceous aerosols from biomass and biofuel burning emissions. J. Geophys. Res.-Atmos., 120, 10285–10296, doi:10.1002/2015JD023697.

Kirchstetter, T. W., T. Novakov, and P. V. Hobbs (2004). Evidence hat the spectral dependence of light absorption by aerosols is affected by organic carbon. J. Geophys. Res., 109, D21208, doi:10.1029/2004JD004999.

Lack, D. A., J. M. Langridge, R. Bahreini, C. D. Cappa, A. M. Middlebrook, and J. P. Schwarz (2012). Brown carbon and internal mixing in biomass burning particles, P. Natl. Acad. Sci. USA, 109, 14802–14807, doi:10.1073/pnas.1206575109.

---

## Author Comment (AC3) · 20 Nov 2017

Author comments
Manuscript: ACP-2017-842
Manuscript title: Investigating biomass burning aerosol morphology using a laser imaging nephelometer

The discussion below includes explanations of changes made to the manuscript by the authors (excluding those in response to reviewers'' comments). All page and line numbers refer to the original manuscript. Added text is indicated by italics, and deleted test is struck through.

1. Recently, the authors became aware of an article on the development of an imaging nephelometer similar to that shown by Dolgos et al. The following has been changed to include this instrument in the discussion of aerosol phase function measurements:

P.3 L.10-12: "The fourth category is the newest and features a *single* pixel array detector *or pair of detectors* with either an elliptical mirror (Curtis et al., 2007) or a wide-angle lens (*Bian et al., 2017;* Dolgos and Martins, 2014) used to image nearly all angles onto the array*(s)*. This type of instrument uses  *one or two detector(s)* with a simple…"

The full reference is:
Bian, Y., Zhao, C., Xu, W., Zhao, G., Tao, J. and Kuang, Y.: Development and validation of a CCD-laser aerosol detective system for measuring the ambient aerosol phase function, Atmos. Meas. Tech., 10(6), 2313–2322, doi:10.5194/amt-10-2313-2017, 2017.

2. We added an additional reference for a new publication to the following:

P.3 L.17: "…visible spectral region (Dolgos and Martins, 2014; *Espinosa et al, 2017*)."

The full reference is:
Espinosa, W. R., Remer, L. A., Dubovik, O., Ziemba, L., Beyersdorf, A., Orozco, D., Schuster, G., Lapyonok, T., Fuertes, D. and Martins, J. V.: Retrievals of aerosol optical and microphysical properties from Imaging Polar Nephelometer scattering measurements, Atmos. Meas. Tech., 10(3), 811–824, doi:10.5194/amt-10-811-2017, 2017.

3. To emphasize that the instrument measures the bulk sample (rather than single particles), we have added the following:

P.4 L.16: "…1 s integration time. *The combination of a wide-angle lens and CCD array allow imaging of the phase function for aerosol particles and gas molecules within the volume of the beam, as discussed in Dolgos and Martins (2014).* The CCD combined…"

4. The $D_f$ value from Gwaze et al. used to model biomass burning fractal-like particles was misreported in our original manuscript as 1.85. The correct value (the average value in Gwaze et al.) is 1.83.

5. The MCE value for Fire A was corrected from 0.946 to 0.949 (P.9 L.20).

6. In the discussion of the importance of fractal morphology for biomass burning aerosol in general, we have now added a sentence acknowledging that past studies indicate coatings condense on BC particles quite quickly in the atmosphere.

P.12 L.9: "…aging in the atmosphere. *Studies have shown that biomass burning aerosol are coated with organic material within hours in the atmosphere (e.g. Akagi et al., 2012; Vakkari et al., 2014). However,* *i*f fractal-like aerosol do not immediately collapse or accumulate sufficient organic coatings to become

spherical, then remote sensing retrievals of wildfire plumes from dry brush or grasses may significantly underestimate the forward scatter from the aerosol. Cheng et al. (2013) showed recently…"

7. The subject heading "Summary and Conclusions" has been moved forward one paragraph, as the discussion of the potential impacts of this study seem to fit better as a summary.

8. We have added the following text regarding the potential importance of accurate phase function algorithms for fresh biomass burning emissions:

P.12 L.14: "…for remote sensing platforms that use reflectance. *This error would likely have been unimportant for past retrievals from MODIS, for example, since cloud masking algorithms often misclassified thick smoke (typically fresh plumes) as clouds (Giglio et al., 2016). However, with improved biomass burning cloud masking algorithms, it will be interesting to see if MODIS retrievals of thick, fresh smoke plumes will be accurate with a spherical morphology algorithm.* This will affect…"

The full reference is:
Giglio, L., Shroeder, W. and Justice, C. O.: The collection of 6 MODIS active fire detection algorithm and fire products, Remote Sens. Environ., 178, 31-41, doi:10.1016/j.rse.2016.02.054, 2016.

9. Figure 2 was changed slightly to make the image more clear:

[Figure]

10. The following reference was changed:

Selimovic, V., Yokelson, R. J., Warneke, C., Roberts, J. M., de Gouw, J.*, Reardon, J.* and Griffith, D. W. T.: Aerosol optical properties and trace gas emissions *by PAX and OP-FTIR for*  laboratory-simulated western US wildfires *during FIREX*, Atmos. Chem. Phys., In *review* , *doi:https://doi.org/10.5194/acp-2017-859,* 2017.